# Knockout of Slo2.2 enhances itch, abolishes $K_{Na}$ current, and increases action potential firing frequency in DRG neurons

Pedro L Martinez-Espinosa[†], Jianping Wu[†‡], Chengtao Yang[†], Vivian Gonzalez-Perez, Huifang Zhou, Hongwu Liang, Xiao-Ming Xia[†], Christopher J Lingle[*]

Department of Anesthesiology, Washington University School of Medicine, St. Louis, United States

**Abstract** Two mammalian genes, *Kcnt1* and *Kcnt2*, encode pore-forming subunits of $Na^+$-dependent $K^+$ ($K_{Na}$) channels. Progress in understanding $K_{Na}$ channels has been hampered by the absence of specific tools and methods for rigorous $K_{Na}$ identification in native cells. Here, we report the genetic disruption of both *Kcnt1* and *Kcnt2*, confirm the loss of Slo2.2 and Slo2.1 protein, respectively, in KO animals, and define tissues enriched in Slo2 expression. Noting the prevalence of Slo2.2 in dorsal root ganglion, we find that KO of Slo2.2, but not Slo2.1, results in enhanced itch and pain responses. In dissociated small diameter DRG neurons, KO of Slo2.2, but not Slo2.1, abolishes $K_{Na}$ current. Utilizing isolectin B4+ neurons, the absence of $K_{Na}$ current results in an increase in action potential (AP) firing and a decrease in AP threshold. Activation of $K_{Na}$ acts as a brake to initiation of the first depolarization-elicited AP with no discernible effect on afterhyperpolarizations.

**\*For correspondence:** clingle@ morpheus.wustl.edu

[†]These authors contributed equally to this work

**Present address:** [‡]Taihe Hospital Neuroscience Research Institute, Hubei University of Medicine, Shiyan, China

**Competing interests:** The authors declare that no competing interests exist.

## Introduction

Potassium channels regulated by cytosolic $Na^+$ ($K_{Na}$) are encoded by two homologous mammalian genes, *Kcnt1* (encoding the Slo2.2 or Slack channel) (*Yuan et al., 2003*) and *Kcnt2* (encoding the Slo2.1 or Slick channel) (*Bhattacharjee et al., 2003*). Recent work has revealed a critical role of $K_{Na}$ channels in neuronal function, through demonstration that several mutations in *Kcnt1* are associated with intellectual disability and childhood epilepsy (*Barcia et al., 2012*; *Heron et al., 2012*; *Martin et al., 2014*). Yet, despite apparently wide-spread expression both in neurons (*Bhattacharjee et al., 2002*, *2005*) and other cells (*Kameyama et al., 1984*; *Niu and Meech, 2000*), the physiological roles of $K_{Na}$ currents during normal patterns of neuronal activity remain poorly understood in part because of the absence of suitably selective pharmacological tools and also the complexities than can arise from manipulations of $Na^+$. Because of potential coupling of $K_{Na}$ activation to $Na^+$ influx through voltage-dependent $Na^+$ (Nav) channels, $K_{Na}$ currents have been proposed to influence repetitive firing (*Yang et al., 2007*; *Gribkoff and Kaczmarek, 2009*) and postexcitatory afterhyperpolarizations (*Franceschetti et al., 2003*; *Gao et al., 2008*). Recently, it has been suggested that $K_{Na}$ currents may be selectively activated by $Na^+$ influx through Nav channel openings that persist at steady state following inactivation (*Hage and Salkoff, 2012*). To further probe the role of $K_{Na}$ currents, we have genetically disrupted *Kcnt1* and *Kcnt2* genes to generate mouse strains in which Slo2.1, Slo2.2, or both subunits together (Slo2 dKO) have been deleted. Because previous work has suggested an important role of Slo2 channels in sensory neurons (*Gao et al., 2008*; *Nuwer et al., 2010*; *Biton et al., 2012*), we examined the consequences of $K_{Na}$ KO on sensory function and dorsal root ganglion (DRG) neuron excitability. The results reveal a role of Slo2.2 channels in acute itch sensation. Pruritic stimuli trigger an immediate increase in itch response in Slo2.2 KO mice, with later time points indistinguishable from WT animals. Furthermore, KO of Slo2.2, but not Slo2.1, removes a $K_{Na}$ current

**eLife digest** The billions of neurons in the brain send information along their lengths in the form of electrical signals called action potentials. These signals are produced by charged ions, such as sodium and potassium ions, moving into and out of the neuron. To 'fire' an action potential, sodium ions rapidly enter the neuron. This produces an electrical spike. Potassium ions then exit the neuron, which causes the electrical activity to subside and allows the neuron to return to a resting state.

The sodium and potassium ions move in and out of the neuron through structures called ion channels. The sodium-activated potassium channels are one type of ion channel; whether these ion channels let potassium ions out of a cell depends on the concentration of sodium ions inside the cell. Slo2.1 and Slo2.2 are two such potassium channels that are present in many different cells, including neurons. Nevertheless, and in spite of how common they are, the exact roles of these channels remain unclear.

Martinez-Espinosa et al. created mice that lack the genes encoding one or both of the Slo2.1 and Slo2.2 ion channels, and compared them with normal mice. Mice that lacked Slo2.2 but not Slo2.1 initially scratched more intensely than normal mice when made to feel an itch, though this increased scratching only occurred briefly. To some extent, the mice that lacked both Slo2 channels also had increased pain sensations.

Martinez-Espinosa et al. observed that in sensory neurons lacking the Slo2.2 sodium-dependent potassium channels, the neurons fired more action potentials. The increase in firing is thought to underlie the enhanced itching and pain sensations.

Taken together, the results suggest that the activity of sodium-activated potassium ion channels makes it less likely for a neuron to fire an action potential. Future work will need to address whether the activity of sodium-activated potassium channels is linked to specific kinds of sodium channels, and why the absence of the sodium-activated potassium current only enhances the immediate response to itch stimuli. The availability of these mice that lack Slo2 subunits provides an important new tool for evaluating the role of sodium-activated channels in other neuronal systems.

from all small-diameter DRG neurons examined. To examine effects of Slo2 KO on DRG excitability, we focused on small diameter neurons, immunoreactive for isolectin B4 (IB4+), which are known to be enriched in neurons responsive to itch and pain stimuli (*Lallemend and Ernfors, 2012*). Slo2 KO increases firing frequency at any level of current injection, while decreasing both rheobase and action potential (AP) threshold. Contrary to the view that $K_{Na}$ current functions primarily during AP repolarization and afterhyperpolarization (*Schwindt et al., 1989*; *Franceschetti et al., 2003*; *Wallen et al., 2007*), we propose that in DRG neurons activation of $K_{Na}$ current precedes AP initiation thereby acting as a brake to AP firing. During completion of this work, another paper describing a Slo2.2 KO mouse (*Lu et al., 2015*) importantly identified a potential role of Slo2.2 in DRG in a neuropathic pain model. Here we reveal a role of Slo2.2 in acute sensory responses and provide a new explanation for how cell firing is altered by Slo2.2 channels.

## Results

### Generation and validation of Slo2.1 and Slo2.2 KO animals

Slo2.1 (gene: *Kcnt2*) and Slo2.2 (gene: Kcnt1) KO mice were generated via homologous recombination of specific targeting DNA fragments (*Figure 1A,D*) into the genome of mouse embryonic stem (ES) cells with confirmation by Southern blot (*Figure 1B,E*), generation of chimeric mice following injection of recombinant ES cells into C57BL/6 blastocysts, and then ultimately Cre/loxP mediated deletion of the targeted exons. Successful incorporation of the mutant allele into mice was confirmed by PCR genotyping of genomic DNA extracted from mouse tails (*Figure 1C,F*). The absence of specific native Slo2 protein was confirmed by western blots of total brain membrane proteins (*Figure 2*; See 'Materials and methods' for discussion of Slo2 epitopes identified by antibodies). Enrichment of brain Slo2 protein via sequential co-immunoprecipitation (co-IP) and western blot further validated the successful KO of Slo2 proteins and also established that Slo2.1 and Slo2.2 coassemble in WT brain (*Figure 2B–E*), as indicated in earlier work (*Chen et al., 2009*). As a

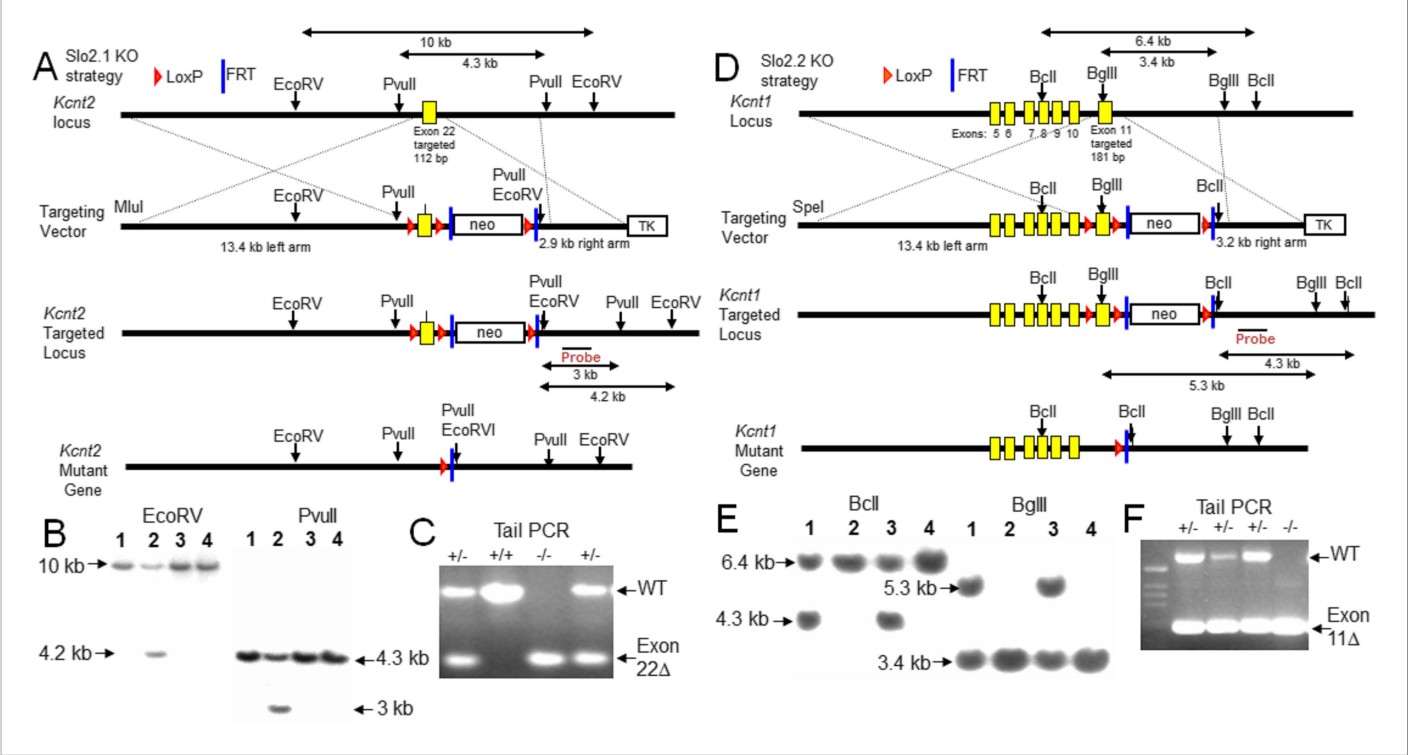

**Figure 1**. Construction and validation of Slo2.1 and Slo2.2 KO mice. (**A**) Upper row: map of WT mouse *Kcnt2* (encoding Slo2.1) gene locus within genomic DNA bracketing the targeted exon 22. Second row: map of the targeting vector, showing M1uI site for vector linearization, targeted exon 22 with a 1.8 kb neomycin gene cassette flanked by LoxP and FRT sites, and a 2.8 kb thymidine kinase (TK) gene cassette. The overall size of the *Kcnt2* genomic DNA for homologous recombination (left arm + right arm) is 16.3 kb. Third row: map of the recombinant allele in targeted embryonic stem (ES) clones following homologous recombination of the *Kcnt2* KO region into the targeted locus. The *neo* gene cassette is eliminated by Flp-FRT mediated deletion. Fourth row: map of the mutant *kcnt* allele following Cre-loxP mediated deletion of the targeted exon. Shown are the elements and restriction enzyme sites used in generation and verification of the targeted mutant allele. Location of the probe used in genomic Southerns for the selection of recombinant ES clones is indicated. After enzyme digestion treatments, the WT allele fragments detected by the probe are 10 kb (by EcoRV) and 4.3 kb (by PvuII), while the recombinant allele fragments detected by the probe are 4.2 kb (by EcoRV) and 3 kb (by PvuII), respectively. (**B**) Genotype analysis of ES cell lines by Southern blot analysis. After enzyme digestion with either EcoRV (left) or PvuII (right), genome DNA obtained from recombinant ES colonies, containing both wild type allele and targeted recombinant allele, shows two corresponding fragments identified by the probe. (**C**) PCR verification of animal genotypes. The target exon is removed by mating heterozygous (HET) F1 mice with early embryonic expression Cre-mice (EIIa-Cre, Jackson). The predicted amplicons are 579 bp for WT and 269 bp for the exon 22 deleted mutant. (**D**) Upper row: map of WT mouse *Kcnt1* (encoding Slo2.2) gene locus bracketing the targeted exon 11. Second row: map of the targeting vector, showing SpeI site for vector linearization, targeted exon 11 and a 1.8 kb neomycin gene cassette flanked by LoxP and FRT sites, and a TK gene cassette. The overall size of *Kcnt1* genomic DNA for homologous recombination is 16.6 kb. Third row: map of the recombinant allele in targeted ES clones following homologous recombination of the *kcnt1* region into the targeted locus. The *neo* gene cassette is then eliminated by Flp-FRT mediated deletion. Fourth row: map of the mutant *Kcnt1* allele following Cre-loxP mediated deletion of the targeted exon. The location of the probes used in genomic Southerns are also indicated. After enzyme digestion treatments, the WT allele fragments detected by the probe are 6.4 kb (by BclI) and 3.4 kb (by BglII), while the recombinant allele fragments detected by the probe are 4.3 kb (by BclI) and 5.3 kb (by BglII) respectively. (**E**) Genotype analysis of ES cell lines by Southern blot analysis. Expected fragment sizes for either BclI (left) or BglII (right) restriction enzyme digestion are shown for both wild type and targeted homologous recombinant. (**F**) PCR verification of *Kcnt1* exon 11 deletion. The target exon is removed by mating HET F1 mice with early embryonic expression Cre-mice (EIIa-Cre, Jackson). The predicted amplicons for WT and the exon 11 deleted mutant were 607 bp and 200 bp, respectively.

guide to tissues of interest for future study, quantitative RT-PCR was employed on various tissues to define the relative abundance of message for *Kcnt1* and *Kcnt2* message (*Figure 2F*). mRNAs encoding either Slo2.1 and Slo2.2 are broadly present in the central nervous system, with message for Slo2.1 notably more abundant in heart and aorta and message for Slo2.2 relatively enriched in other tissues including DRG and cerebellum. The selective expression of transcript for Slo2.1 in rat heart has been previously reported (*Bhattacharjee et al., 2003*). Based on the RT-PCR results, we examined DRG, spinal cord, cortex, cerebellum and heart for the presence of Slo2.1 and Slo2.2 subunits using sequential IP and western blot (*Figure 2G–J*). Slo2.1 protein was detected in DRG, spinal cord, cortex

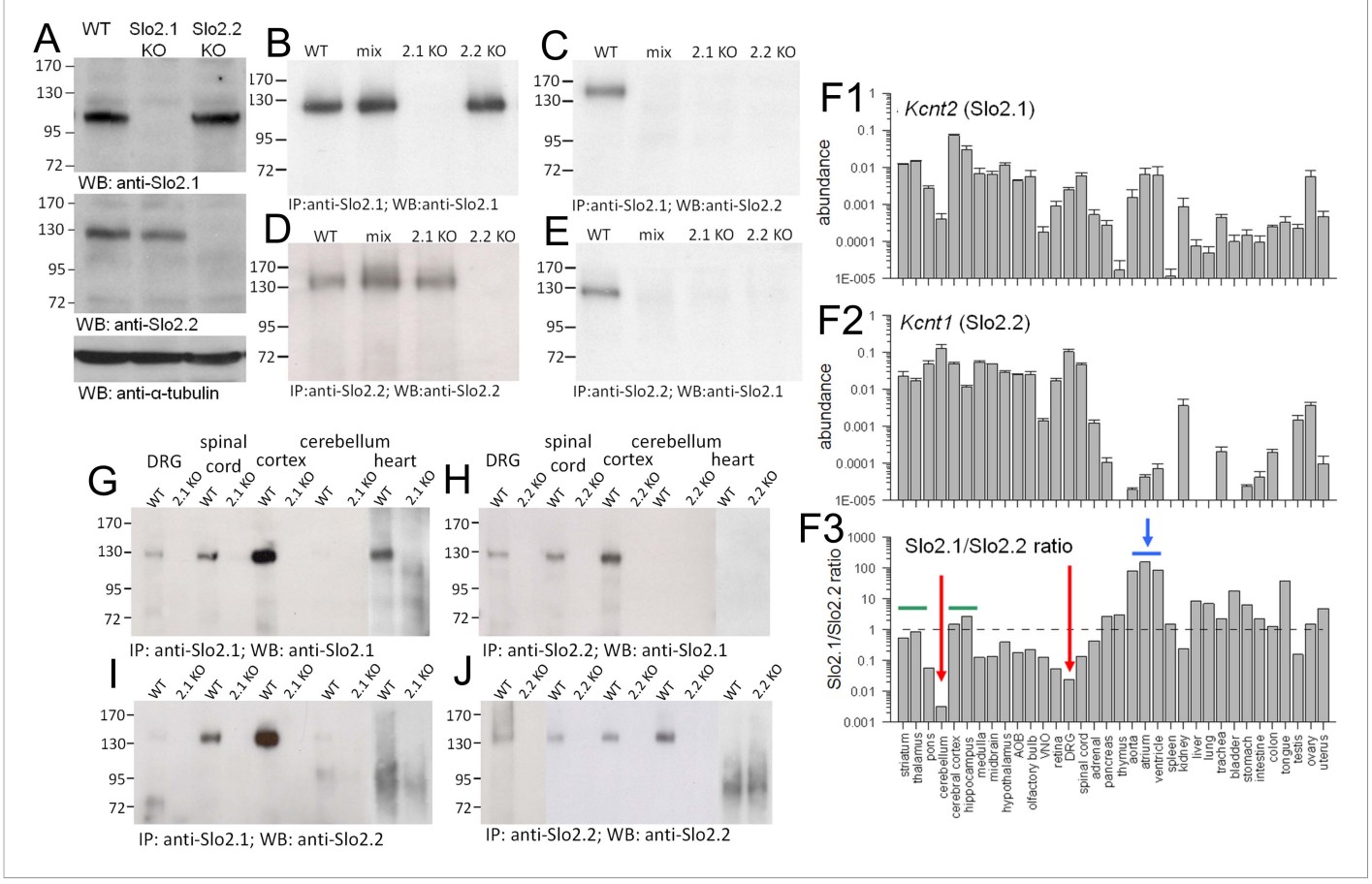

**Figure 2**. Slo2.1 and Slo2.2 subunits are absent in *Kcnt2* and *Kcnt1* KO mice, respectively, exhibit differential tissue distribution, and coassemble in some tissue. (**A**) Top, brain membrane proteins from WT, Slo2.1 KO, and Slo2.2 KO mice were probed with N11/33 anti-Slo2.1 antibody (Antibodies Inc.). Middle, brain membrane proteins were separated and probed with N3/26 anti-Slo2.2 mAb (Antibodies, Inc). Slo2.2 protein is absent in Slo2.2 KO mice. No native Slo2.2 protein is present in the Slo2 KO mice, but is found in Slo2.1 KO mice. Bottom, α-tubulin loaded in each lane was probed with anti-α-tubulin Ab. 15 µg of whole brain membrane proteins were loaded in each lane. (**B**) Slo2.1 Ab pulls down Slo2.1 protein from brain membrane proteins in WT and Slo2 KO mice, but not from Slo2.1 KO mice. Anti-Slo2.1 Ab also pulls down Slo2.1 from proteins following mixing of separate Slo2.2 KO and Slo2.1 KO membrane preparations (mix). 25 µg of whole brain proteins were subjected to IP procedures and the IP products were loaded in each lane. (**C**) IP with anti-Slo2.1 Ab pulls down Slo2.2 only in WT membrane proteins, but not in mixed proteins, or membrane proteins from Slo2.1 KO or Slo2.2 KO mice. 62.5 µg of whole brain proteins were subjected to IP procedures with the IP products loaded in each lane. (**D**) Following IP with anti-Slo2.2, Slo2.2 is detected in proteins from WT, mixed, and Slo2.1 KO membranes. 25 µg of whole brain proteins were subjected to IP procedures and the products loaded in each lane. (**E**) IP with anti-Slo2.2 Ab pulls down Slo2.1 only from WT membrane proteins. 62.5 µg of whole brain proteins was subjected to IP procedures and the products loaded in each lane. (**F1**) Abundance of message for Slo2.1 relative to β-actin message is plotted for various tissues. Here and in (**F2**), message was measured in triplicate from each of three mice. (**F2**) Slo2.2 message abundance is plotted. (**F3**) The ratio of message for Slo2.1 to Slo2.2 measured by quantitative rt-PCR is shown for various tissues. Dotted line indicates approximately equimolar RNA amounts. Red arrows highlight enrichment of Slo2.2 message. Horizontal blue bar and arrow highlight relative enrichment of message for Slo2.2 in heart tissues. (**G**) IP with anti-Slo2.1 shows presence of Slo2.1 protein in DRG, spinal cord, cortex and heart, but not cerebellum. Protein amounts used in IPs were: DRG, 3 mg; spinal cord, 1 mg; cortex, 0.3 mg; cerebellum; 2 mg; heart, 30 mg. (**H**) IP with anti-Slo2.2 pulls down Slo2.1 in DRG, spinal cord and cortex, but not in cerebellum and heart. Protein amounts used in IPs were: DRG, 3 mg; spinal cord, 1 mg; cortex, 0.5 mg; cerebellum; 0.25 mg; heart, 30 mg. (**I**) IP with anti-Slo2.1 pulls down Slo2.2 in spinal cord and cortex. (**J**) IP with anti-Slo2.2 shows presence of Slo2.2 in all tested tissues except heart. Western blots were repeated three times in all cases, except twice for DRG.

and heart, but only a very weak band was seen from cerebellum (*Figure 2G*). Slo2.2 was observed in DRG, spinal cord, cortex, and cerebellum, but not detectable in heart (*Figure 2J*). Co-IP between Slo2.1 and Slo2.2 was observed in those tissues for which both subunits were detectable: DRG, spinal cord, and cortex (*Figure 2H,I*). Because K_Na currents have been described in sensory neurons (*Gao et al., 2008*; *Tamsett et al., 2009*; *Nuwer et al., 2010*), we chose DRG as a convenient system for investigation of potential physiological roles.

## Slo2.2, but not Slo2.1, KO mice exhibit an enhanced response to pruritic stimuli

WT and Slo2 KO mouse strains were evaluated with various tests of sensory function. In a 55℃ hotplate test, single KO of either Slo2.1 or Slo2.2 did not influence the response latency, although Slo2 dKO mice exhibited a briefer latency than WT mice (*Figure 3A*). In a formalin test, no differences were observed between WT and Slo2 dKO mice (*Figure 3B*). The absence of a difference in hotplate or formalin response in Slo2.2 KO mice agrees with recent observations on another Slo2.2 KO mouse (*Lu et al., 2015*).

Hindpaw injection of capsaicin elicits a characteristic licking behavior which was somewhat enhanced in Slo2 dKO mice (*Figure 3C*). Because the intensity of a sensory stimulus may affect whether $K_{Na}$ currents influence sensory function, we compared responses to a series of capsaicin doses (*Figure 3D*). Consistent with this idea, pronounced differences between WT and Slo2 dKO mice were present at doses in excess of 0.0001 µg up through 0.01 µg, with weaker differences at 0.03, 0.1 µg, and higher concentrations. These results indicate that mice lacking both Slo2.1 and Slo2.2 channels exhibit an enhanced aversion to moderate doses of capsaicin and that Slo2 dKO can influence the acute response to sensory stimuli.

We next tested several pruritic compounds in a standard itch assay (*Sun and Chen, 2007*). Chloroquine (CQ, *Figure 4A–F*), histamine (HA, *Figure 4G–L*, *Figure 4—figure supplement 1A–E*), and compound 48–80 (*Figure 4—figure supplement 1F*) elicited robust enhancement of scratching behavior in Slo2 dKO mice (*Video 1* for the case of CQ), but not WT mice (*Video 2*), during the first 5 min following injection. No difference in itch behavior was observed between WT and Slo2 dKO mice after the first 5 min. KO of only Slo2.2 also revealed a similar alteration in the itch phenotype during the first 5 min after injection (CQ: *Figure 4C,D*; HA: *Figure 4I,J*). The enhanced itch was also observed in heterozygous Slo2.2 mice. In contrast, WT and Slo2.1 mice exhibited no difference in response to either CQ (*Figure 4E,F*) or HA (*Figure 4K,L*).

Because the time course of the early itch response was similar to capsaicin responses (*Figure 3C*), it seemed possible that pruritic stimuli in the Slo2 KO mice were perceived as something distinct from itch. A cheek injection assay has been proposed to distinguish itch from pain (*Shimada and LaMotte, 2008*). In the cheek, injection of HA elicits hindlimb scratching, while capsaicin injection elicits forepaw wiping (*Shimada and LaMotte, 2008*), suggesting that they are being perceived differently. We wondered whether a pruritic stimulus injected into the cheek of a Slo2 dKO mouse might elicit a capsaicin-like forepaw wiping response. In our hands, cheek injection of CQ in WT animals was associated with two types of behaviors, hindlimb scratching of the injected site, but also some forepaw wiping presumably reflecting grooming (*Figure 5*). In the dKO animals, forepaw wiping was no different than in WT (*Figure 5B*), but the hindlimb scratching was markedly increased only during the first 5 min (*Figure 5A*). Whatever the basis of the enhanced response to cheek injection of CQ in Slo2 dKO mice, the response is characteristic of pruritic stimuli and not of capsaicin.

## $K_{Na}$ current is absent in DRG neurons from Slo2.2, but not Slo2.1, KO mice

Sensory neurons contain a rich variety of $K^+$ currents (*Vydyanathan et al., 2005*; *Dobler et al., 2007*; *Li et al., 2007*; *Cho et al., 2009*; *Zhang et al., 2010b*; *Liu et al., 2013*) that complicate unambiguous definition of $K_{Na}$ current, for which selective pharmacological tools are lacking. We have not had reliable success with subtractive methods involving $Na^+$ current inhibition or $Na^+$ replacement. To test for the presence of $K_{Na}$ current in small diameter DRG neurons, we used a method previously applied to rat DRG neurons (*Bischoff et al., 1998*): a $K^+$ background current arising from defined pipette $Na^+$ is measured using hyperpolarizing voltage-steps during the first 5 min following formation of the whole-cell recording configuration. With 0 mM pipette $Na^+$, little background current is observed with voltage-steps from −80 to −120 mV (*Figure 6A,C*). With 70 mM pipette $Na^+$, net current elicited by the same voltage-step gradually increases over 3 min reaching a plateau near 1 nA (*Figure 6A,C*). At longer times following whole-cell access, current activated by 70 mM pipette $Na^+$ gradually diminishes (*Figure 6—figure supplement 1*) despite no change in voltage-dependent $Na^+$ current. As in rat DRG neurons (*Bischoff et al., 1998*), the $K_{Na}$ current is blocked by extracellular 20 mM $Cs^+$, with stronger inhibition at −120 mV than −80 mV reflecting the voltage-dependence of $Cs^+$ inhibition

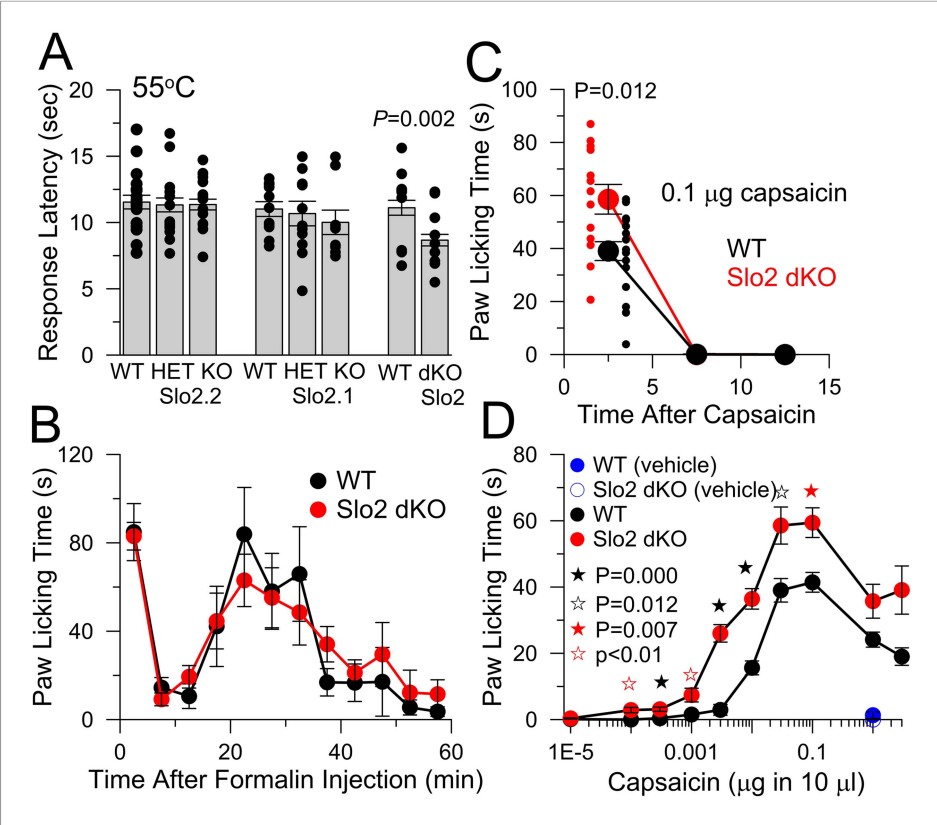

**Figure 3**. Slo2 dKO shortens hotplate response latency, increases responses to hindpaw injections of capsaicin, but does not influence formalin responses. (**A**) Latencies to aversive response following placement on a 55°C hotplate are plotted for the indicated genotypes, showing means, sem, and individual latencies. From left to right, n = 19, 19, 24, 11, 13, 9, 17, and 24. Only in the WT vs Slo2 dKO comparison was a difference noted (p = 0.002; KS test). (**B**) Following formalin injection, time spent in licking the hindpaw was determined for 5 min intervals for WT (n = 10) and Slo2 dKO (n = 9) mice. Here and below, behavioral tests over time display measurements centered in each 5 min interval. (**C**) Time course of licking response to hindpaw injection of 0.1 µg capsaicin. Small symbols, individual mice. p = 0.012 (KS test). Vehicle: 10 µl volume with 0.35% EtOH. (**D**) Time spent licking was determined over 10 min following hindpaw injections of the indicated capsaicin quantities in 10 µl vehicle for WT (n = 9, 9, 20, 20, 18, 20, 20, 20, 20, and 9 from low to high capsaicin) and Slo2 dKO (n = 10, 10, 11, 9, 14, 26, 13, 18, 10, and 10) genotypes. Vehicle alone was without effect (n = 10 for both WT and Slo2 dKO). For filled black, open black, and filled red stars, p values correspond to KS statistic with p = 0.000 (filled black stars), p = 0.007 (filled red stars), and p = 0.012 for open black star. For open red stars, a t-test statistic was used with p < 0.01. Highest capsaicin concentrations showed no difference between WT and Slo2 dKO mice.

(*Figure 6A*, *Figure 6—figure supplement 2*). The average amplitude of $K_{Na}$ current was similar for WT and Slo2.1 KO DRG neurons (*Figure 6B,C*), while there was no $K_{Na}$ current in Slo2.2 KO or Slo2 dKO neurons (*Figure 6B,C*). Despite considerable variability in total $K_{Na}$ current among neurons from either WT or Slo2.1 KO animals (*Figure 6D*), the total current always exceeds that observed in WT cells with 0 $Na^+$, or in Slo2 dKO or Slo2.2 KO cells with 70 mM $Na^+$ (*Figure 6D*). Excised inside-out patches confirmed that Slo2 dKO removed a Na–dependent $K^+$ channel (*Figure 6E*) which exhibited little voltage-dependence over the range of −80 through −20 mV (*Figure 6E*, *Figure 6—figure supplement 3A*) with a single channel conductance of about 127 pS (*Figure 6E*, *Figure 6—figure supplement 3B*). Finally, we compared the whole-cell steady-state current–voltage (I–V) relationship between WT and Slo2 dKO cells over the range of −125 to −25 mV, with 70 mM pipette $Na^+$ along with the steady-state IV relationship persisting in WT cells after 30 min with 70 mM pipette $Na^+$ (*Figure 6F*). This shows the relatively voltage-independent nature of the background $K_{Na}$ conductance (reversal at $E_K$) when the cytosolic $Na^+$ concentration is constant.

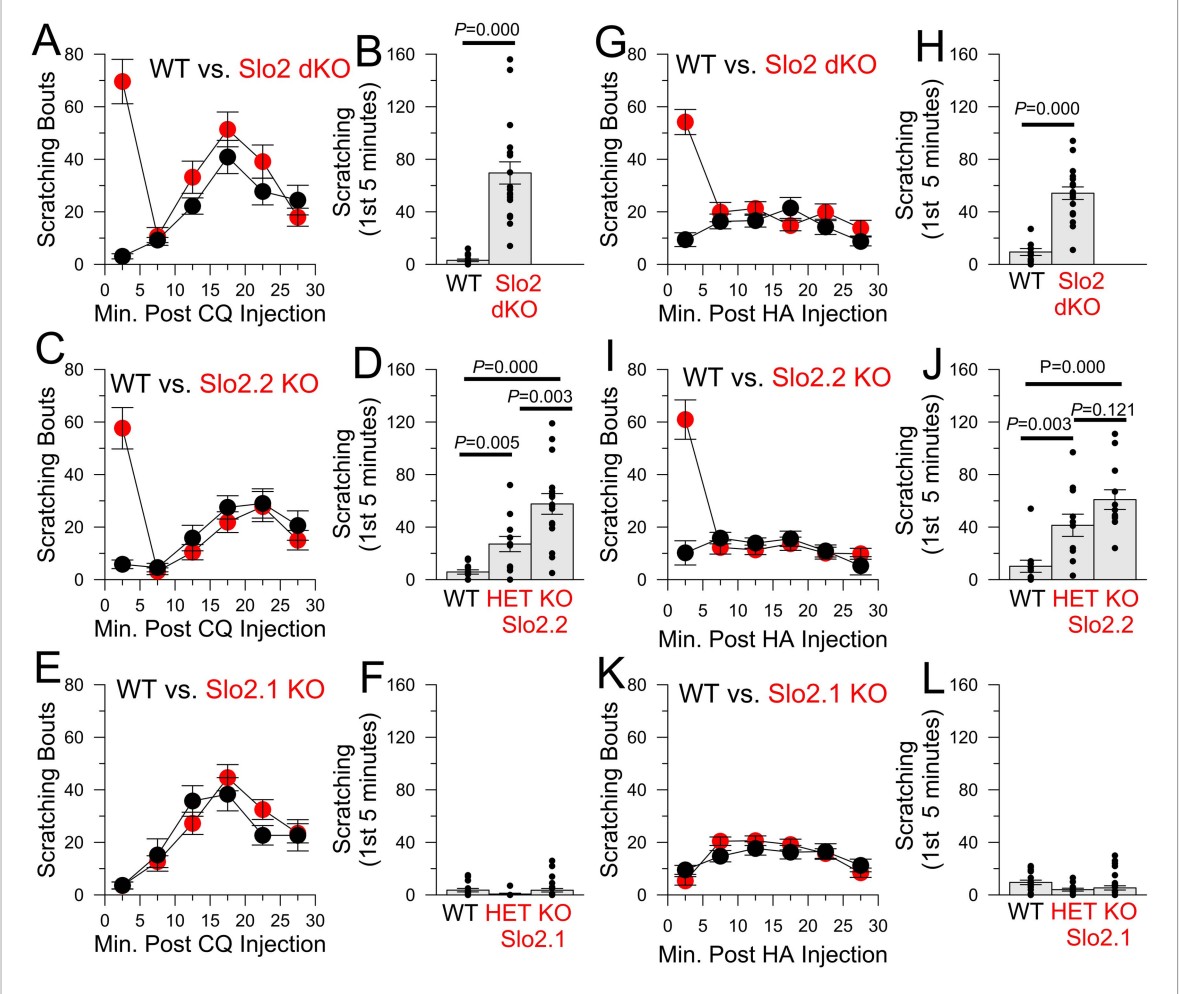

**Figure 4**. The absence of Slo2.2, but not Slo2.1, results in enhancement of chloroquine (CQ) and histamine (HA)-induced itch. (**A**) Each point shows mean number of scratching bouts per 5 min bins for WT mice (n = 15, black circles) and Slo2 dKO mice (n = 19, red circles) after injection of 200 µg CQ. (**B**) Mean scratching bouts during first 5 min are summarized for WT and Slo2 dKO mice from (**A**), along with determinations from individual mice (circles). Over the first 5 min, WT and Slo2 dKO mice differ at p = 0.000 (KS-test). (**C**) Slo2.2 KO mice exhibit enhanced responsiveness to CQ injection. (**D**) Mean scratching bouts during the first 5 min after CQ injection for WT (n = 12), Slo2 HET mice (n = 12) and Slo2.2 KO mice (n = 16). KS-test comparisons: WT vs Slo2.2 HET, p = 0.005; WT vs Slo2.2 KO, p = 0.000; Slo2.2 HET vs Slo2.2 KO, p = 0.003. (**E**) Slo2.1 KO mice exhibit CQ responsiveness identical to WT mice. (**F**) Mean scratching during the first 5 min after CQ injection for WT (n = 16), Slo2.1 HET (n = 11) and Slo2.1 KO (n = 16) mice. (**G**) Responses of WT (n = 15) and Slo2 dKO (n = 18) mice following injection of 1 mg HA. (**H**) Scratching during first 5 min following HA injection for WT and Slo2 dKO mice. Over the first 5 min, WT and Slo2 dKO mice differ at p = 0.000. (**I**) HA-induced scratching behavior for WT and Slo2.2 KO mice. (**J**) Mean and individual values of scratching during first 5 min for WT (n = 11), Slo2.2 HET (n = 11), and Slo2.2 KO (n = 12) mice. KS-test comparisons: WT vs Slo2.2 HET, p = 0.003; WT vs Slo2.2 KO, p = 0.000; Slo2.2 HET vs Slo2.2 KO, p = 0.121. (**K**) HA-induced scratching behavior for WT and Slo2.1 KO mice. (**L**) Mean and individual values of scratching during first 5 min for WT (n = 19), Slo2.1 HET (n = 13), and Slo2.1 KO (n = 30) mice.

The following figure supplement is available for figure 4:

**Figure supplement 1**. Concentration-dependence of itch response to HA and compound 48–80.

In WT DRG neurons, the average $K_{Na}$ background current with 70 mM pipette $Na^+$ in IB4+ neurons did not differ significantly from that in IB4− neurons (*Figure 6—figure supplement 4A*). Our results suggest that essentially all dissociated small diameter DRG neurons express $K_{Na}$ current which can be attributed exclusively to Slo2.2 subunits. The magnitude of the $K_{Na}$ current decreased with time in culture, being ~1013 ± 95 pA (n = 44 cells) after 2–10 hr of culture, but only 277 ± 70 pA (n = 9 cells) after 2–3 days in culture (*Figure 6—figure supplement 4B*). Measurement of $K_{Na}$ current in a set of

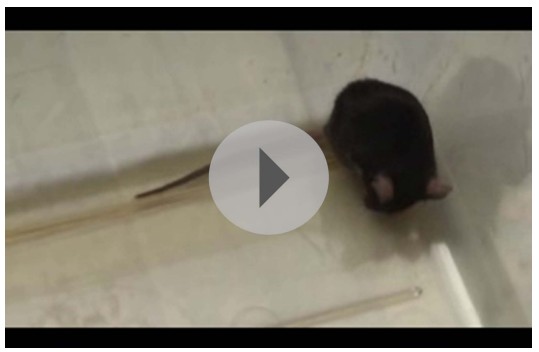

**Video 1.** Response of a Slo2 dKO mouse to CQ injection (related to *Figure 4A*). The nape of the neck of a Slo2 dKO mouse was injected with 10 µl 200 µM choroquine. Video recording was begun about 10 s after injection.

small diameter neurons in DRG slices yielded similar results (*Figure 6—figure supplement 5*). However, one difference was that, although Slo2.2 accounted for most of the $K_{Na}$ current in DRG neurons in slices, after 2–3 min of dialysis of the pipette solution into the Slo2.2 KO neurons, some residual $Na^+$-dependent current was observed. Although neurons in slices may be uniquely affected by dialysis of 70 mM cytosolic $Na^+$, this observation raises the possibility that channels containing Slo2.1 subunits may be present at more peripheral locations in the DRG neurons, perhaps consistent with the presence of message encoding Slo2.1 and some Slo2.1 protein in DRG samples, as described above.

## DRG neurons from Slo2 dKO mice exhibit increased excitability and reduced rheobase

Small diameter DRG neurons exhibit a complex range of electrical properties reflecting a rich variety of Nav (*Vijayaragavan et al., 2001*; *Ho and O'Leary, 2011*) and Kv channels (*Zhang et al., 2010b*). Such neurons are also heterogeneous (*Petruska et al., 2000*; *Dirajlal et al., 2003*) in regards to sensitivity to various chemical signals. Given the presence of Slo2.2-dependent $K_{Na}$ current in all DRG neurons we sampled, $K_{Na}$ currents may influence excitability in several different classes of neurons. Since tests for phenotypic consequences of Slo2.2 KO pointed to neurons involved in itch and, to a lesser extent, pain, we limited our analysis to small-diameter IB4+ neurons, likely to be enriched in neurons involved in itch and polymodal pain sensation (*Lallemend and Ernfors, 2012*).

Neurons were selected for recordings based on size defined from membrane capacitance (WT: 16.1 ± 0.3 pF [±sem; n = 64]; Slo2 dKO: 15.9 ± 0.5; [n = 41]) and the presence of IB4 reactivity (*Dirajlal et al., 2003*). Furthermore, neurons were prepared from 3 to 5 week old mice to help ensure relative numbers of IB4+ and Ret-expressing neurons (*Molliver et al., 1997*) more consistent with acquisition of adult itch and polymodal pain-sensing (*Lallemend and Ernfors, 2012*). We used Slo2 dKO neurons to guarantee complete absence of any Slo2-dependent $K_{Na}$ current.

A 1 s current step to different amplitudes was used to compare numbers of evoked APs in both WT and Slo2 dKO neurons with either 10 mM (*Figure 7A*) or 0 mM pipette $Na^+$ (*Figure 7B*). Average resting potential ($V_m$) was adjusted to –60 mV, prior to the depolarizing current pulses. Despite considerable variability in the maximum firing rates among both WT and Slo2 dKO neurons, AP firing was, on average, more robustly elevated in Slo2 dKO neurons than in WT neurons for identical amounts of injected current (*Figure 7C,D*). The increase in firing in Slo2 dKO neurons was observed at all levels of current injection, both with 10 and 0 mM pipette $Na^+$ (*Figure 7E–G*). AP firing did not differ between 10 and 0 mM pipette $Na^+$ within WT neurons or within Slo2 dKO neurons. The increase in AP firing associated with $K_{Na}$ loss is consistent with increased AP firing of embryonic (E15) rat DRG neurons following protein kinase A-mediated internalization of $K_{Na}$ channels (*Nuwer et al., 2010*).

Standard protocols were used to compare basic electrical properties of WT and Slo2 dKO neurons either with 10 mM (*Table 1*, top) or 0 mM (*Table 1*, bottom) pipette $Na^+$. To measure

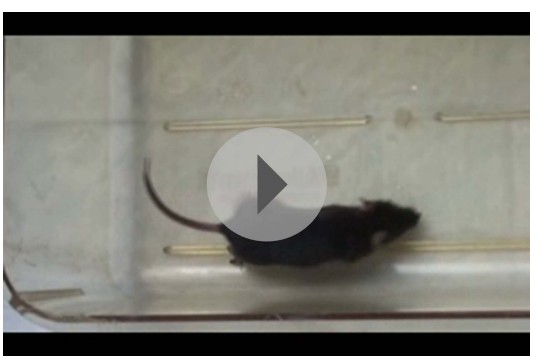

**Video 2.** Response of a WT mouse to CQ injection (related to *Figure 4A*). The nape of the neck of a WT mouse was injected with 10 µl 200 µM choroquine. Acquisition of video was begun about 10 s after injection.

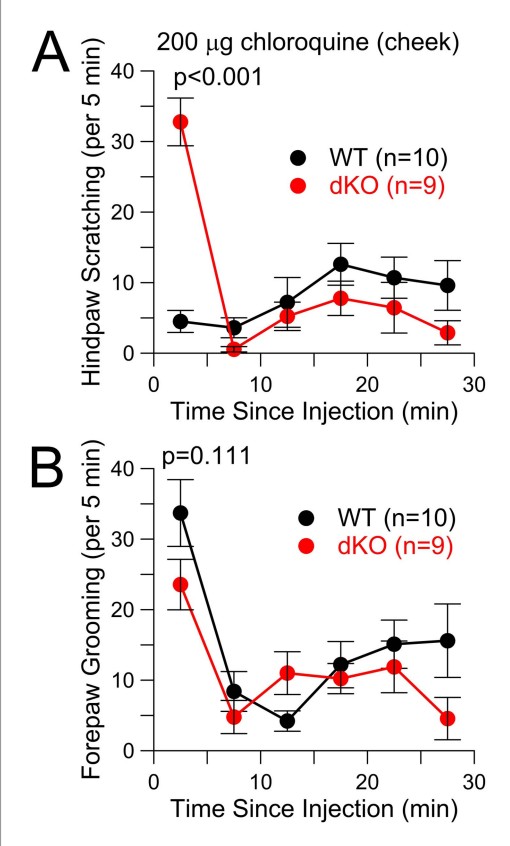

**Figure 5**. CQ enhances itch-type behavior following cheek injection, but not pain-type behavior. (**A**) Total scratching bouts using the hindpaw to scratch the cheek was monitored following cheek injection of 200 µg CQ in WT and Slo2 dKO mice. During the first 5 min interval, distributions differed at p < 0.001 (Student's t-test). (**B**) Bouts of forepaw grooming were monitored following CQ cheek injection for WT and Slo2 dKO mice. There was no difference in the first 5 min.

rheobase, 20 ms depolarizing current injections were applied from a −60 mV holding potential either with 10 (*Figure 8A*) or 0 mM pipette $Na^+$ (*Figure 8B*). This defines a minimal amount of injected current necessary to elicit an AP. Despite considerable variance within both WT and dKO cells, both at 10 and 0 mM $Na^+$ less current was required to elicit an AP in the dKO neurons (*Figure 8C*; *Table 1*). This difference between WT and dKO neurons suggests that $K_{Na}$ is activated prior to or during the weak depolarizations that begin to elicit Nav activation and is not influenced by pipette $Na^+$ over the range of 0–10 mM.

We next compared the properties of single APs in WT and Slo2 dKO neurons elicited by a single 20 ms 100 pA current injection with 0 mM pipette $Na^+$ (*Figure 8D*), a stimulus usually sufficient to evoke an AP in both WT and dKO cells. AP waveforms were then transformed into phase plots (dV/dt vs V) for each cell (*Figure 8E*). Since peak dV/dt can vary substantially among cells, we have defined the threshold in a given cell as the $V_m$ value at which dV/dt reaches 10% of its peak value (dotted lines on *Figure 8E,F*). This comparison shows that APs are initiated from a more negative $V_m$ in dKO cells than in WT cells (*Figure 8G*). In contrast to the effects of Slo2 dKO on AP initiation, a number of other properties of single APs, including peak AP amplitude, AP half-width, and AP after-hyperpolarization, showed no obvious differences (*Table 1*). However, with both 10 and 0 mM pipette $Na^+$, dKO cells exhibited a somewhat more depolarized $V_m$, although no obvious difference in input resistance ($R_{in}$) measured from a 10 mV step from −60 to −70 mV was noted. Potential reasons for the apparent discrepancy between $V_m$ and $R_{in}$ will be considered below.

If the difference in apparent AP threshold between WT and dKO cells arises from outward $K_{Na}$ current present in the WT cells that delays the activation of Nav current, a voltage-clamp ramp protocol that better approximates the slow depolarization preceding an AP might also reveal a difference between WT and dKO neurons. From a holding potential of −60 mV, cells were therefore stimulated with a 40 ms voltage-ramp up to −20 mV (*Figure 8H*). We observed that the $V_m$ at which the overall current became net inward was more negative in dKO cells compared to WT cells (*Figure 8H,I*). Prior to the surge of Nav current activation, the ramp reveals a modest outward current, which is larger on average in WT cells and which in WT cells slightly shifts rightward the voltage at which net current becomes inward, relative to dKO cells.

Although the properties of the ramp-activated outward current and shift in 0 current potential are generally consistent with the loss of outward current activated at the onset of depolarization, a concern in regards to the above experiments is that the comparisons are being made between cell populations from genotypically distinct animals. For example, a shift in Nav channel activation to more negative potentials in dKO neurons might produce qualitatively similar effects. We therefore tested several inhibitors and activators of $K_{Na}$ current as tools to examine the properties of ramp-activated current in WT cells, but slow onset of action and non-specific effects on other ion channels precluded their use. As an alternative, having shown that extracellular $Cs^+$ inhibits $K_{Na}$ current, we examined the

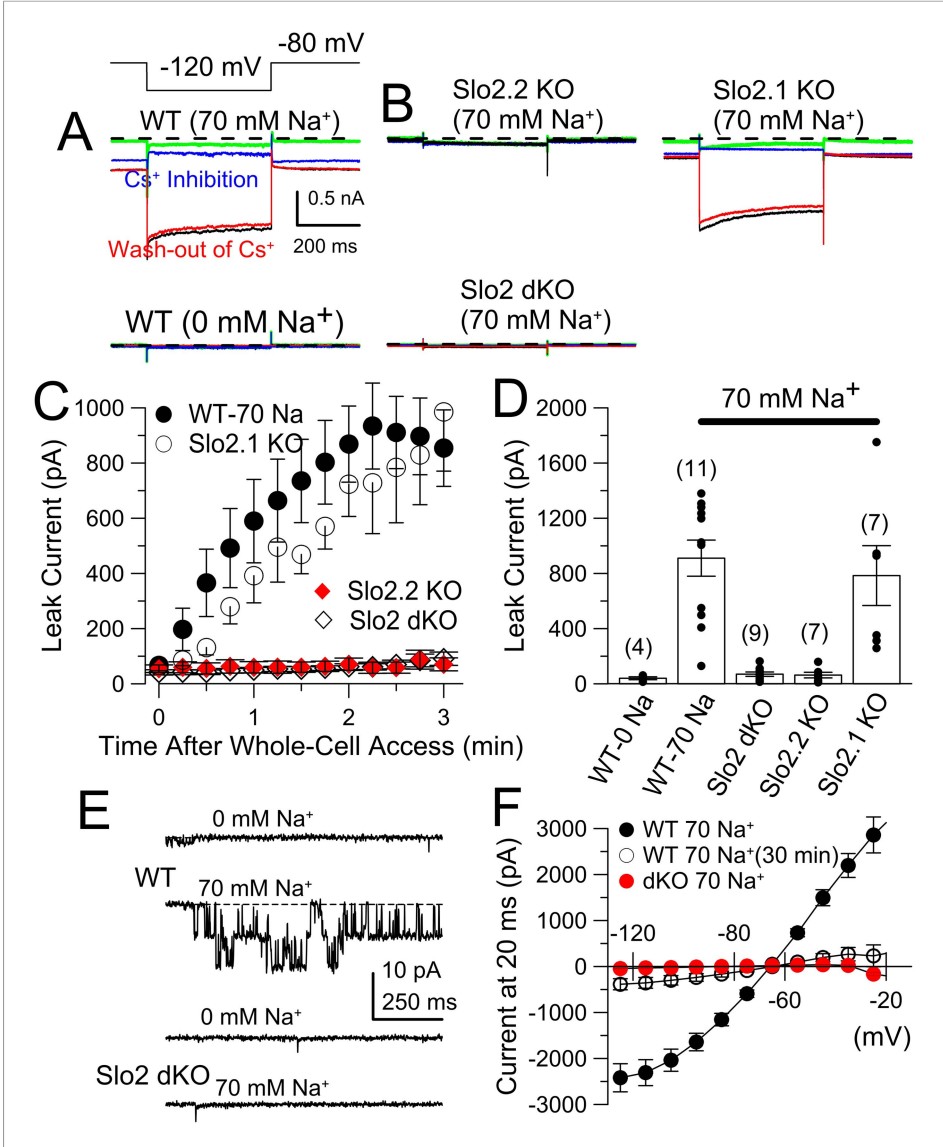

**Figure 6**. The absence of Slo2.2 reduces Na$^+$-dependent leak current in acutely dissociated mouse DRG neurons. (**A**) Traces on the top show currents (evoked by indicated voltage protocol) for four time points following formation of a whole-cell recording with 70 mM pipette Na$^+$. Green: immediately following whole-cell access; black: 3 min following access; blue: following application of 20 mM Cs$^+$; red: washout of Cs$^+$. On the bottom, traces are from another WT neuron examined with the same procedure, but with 0 mM pipette Na$^+$. (**B**) Panels correspond to the same sequence as shown in (**A**) for a Slo2.2 KO neuron (top left), a Slo2.1 KO neuron (top right), and a Slo2 dKO neuron (bottom). (**C**) The time courses of increases in net current evoked by steps from −80 to −120 mV are shown for WT and the three indicated Slo2 genotypes. (**D**) Mean estimates of leak current and standard errors measured 3 min following whole-cell access are plotted for different test conditions. Circles correspond to individual cells. t-test comparisons yielded: for 0 mM Na$^+$ WT vs 70 mM Na$^+$ WT, p = 0.0015; for 70 mM WT vs Slo2 dKO, p < 0.001; for 70 WT vs Slo2.2 KO, p < 0.001; for Slo2.1 KO vs Slo2.2 KO, p = 0.0038. All other comparisons were p > 0.1. (**E**) Traces on the top show channel activity in a patch excised from a WT DRG neuron bathed either with 0 mM Na$^+$ or 70 mM Na$^+$. Bottom: a similar patch from a Slo2 dKO neuron reveals no channels activated by Na$^+$. (**F**) Voltage-step protocols over the range of −125 mV to −25 mV were used to compare steady-state conductance (measured at the end of a 20 ms command step) (*Figure 6—figure supplement 1*) in WT and dKO neurons with 70 mM pipette Na$^+$, along with WT neurons with 70 mM Na$^+$ after 30 min of recording.

*Figure 6. continued on next page*

*Figure 6. Continued*

The following figure supplements are available for figure 6:

**Figure supplement 1**. $K_{Na}$ current runs down during constant cytosolic 70 mM $Na^+$.

**Figure supplement 2**. $Cs^+$ inhibition of $K_{Na}$ current exhibits voltage-dependence.

**Figure supplement 3**. Confirmation of properties of single $K_{Na}$ channels that are deleted by Slo2 dKO.

**Figure supplement 4**. $Na^+$-dependent leak current is present in both IB4+ and IB4− neurons and runs down with time in culture.

**Figure supplement 5**. The absence of Slo2.2 and, to a lesser extent, Slo2.1, reduces $Na^+$-dependent leak current in mouse DRG neurons in DRG tissue slices.

ability of 20 mM $Cs^+$ to influence excitability and ramp-activated currents in both WT and dKO DRG neurons with 0 mM pipette $Na^+$ (*Figure 9*). As shown above, the voltage-ramp activated a much more pronounced low-voltage outward current in WT cells than in dKO cells, with a marked shift in the 0 current potential (*Figure 9A*). Application of 20 mM extracellular $Cs^+$ to WT neurons also resulted in a reduction in ramp-activated outward current and a shift in the 0 current voltage (p = 0.000; *Figure 9B*), quite comparable to the current observed in the dKO neurons (*Figure 9C*). In contrast, application of 20 mM $Cs^+$ to the dKO cells produced only small shifts in the 0 current voltage (p = 0.675; *Figure 9D*). Overall, 20 mM $Cs^+$ mimicked the effect of Slo2 dKO on the 0 current voltage (*Figure 9E*), while also producing essentially identical effects on measurement of rheobase in the same set of neurons (*Figure 9F*). That an apparent shift in the voltages over which the surge of inward current is observed can occur from $K^+$ channel inhibition is also highlighted in comparisons of the normalized ramp activated currents (*Figure 9—figure supplement 1*), which clearly shows the ability of $Cs^+$ to produce a shift in apparent inward current activation in WT cells which is much more reduced in the dKO cells. Inhibition by $Cs^+$ is likely to differ from Slo2 dKO in two primary ways: first, $K_{Na}$ will not be inhibited completely by $Cs^+$ at these voltages and second, $Cs^+$ is likely to inhibit other $K^+$ currents in addition to $K_{Na}$. However, the results clearly support the view that an apparent shift in inward current activation occurs with inhibition of subthreshold $K^+$ currents, likely to include $K_{Na}$.

The same set of cells was also examined with standard voltage-step protocols to ascertain the properties of peak inward and steady-state outward current (*Figure 9—figure supplement 2*). Step-activated inactivating current and the voltage of half-activation of the inward current was roughly similar in both WT and dKO neurons, with similar reductions produced by 20 mM $Cs^+$ (*Figure 9—figure supplement 2A–D*). The absence of obvious differences between the Nav currents in WT and dKO neurons make it highly unlikely that a difference in Nav current between WT and dKO neurons accounts for the differences in rheobase and ramp-activated 0 current potential. Furthermore, if the effect of Cs on ramp-activated current were to arise from an effect on Nav current, an inhibition of Nav current would be expected to shift the 0 current voltage rightward. This is not observed. Together, these results support the view that the $Cs^+$ induced inhibition of ramp-activated outward current and the shift in the 0 current voltage arise solely from inhibition of a $K^+$ current. Whatever this current is, it is apparently absent in the dKO neurons. Although it is perhaps possible that some other low voltage activated $K^+$ current other than $K_{Na}$ is also absent in the dKO neurons, the simpler view is that the difference in excitability between the WT and dKO neurons arises from the absence of the $K_{Na}$ current itself.

To ascertain whether there might be changes in other components of current between this particular set of WT and dKO neurons, we also compared steady-state current at the end of a 20 ms voltage-step in the same set of cells (*Figure 9—figure supplement 2E,F*). Although net outward current was generally similar in both groups, the dKO cells exhibited a larger outward at command potentials from −10 mV and more positive (*Figure 9—figure supplement 2E*). However, over the range of −120 to almost −20 mV, there was no obvious difference in this steady-state current (*Figure 9—figure supplement 2F*). With 0 mM pipette $Na^+$, no significant difference was observed in

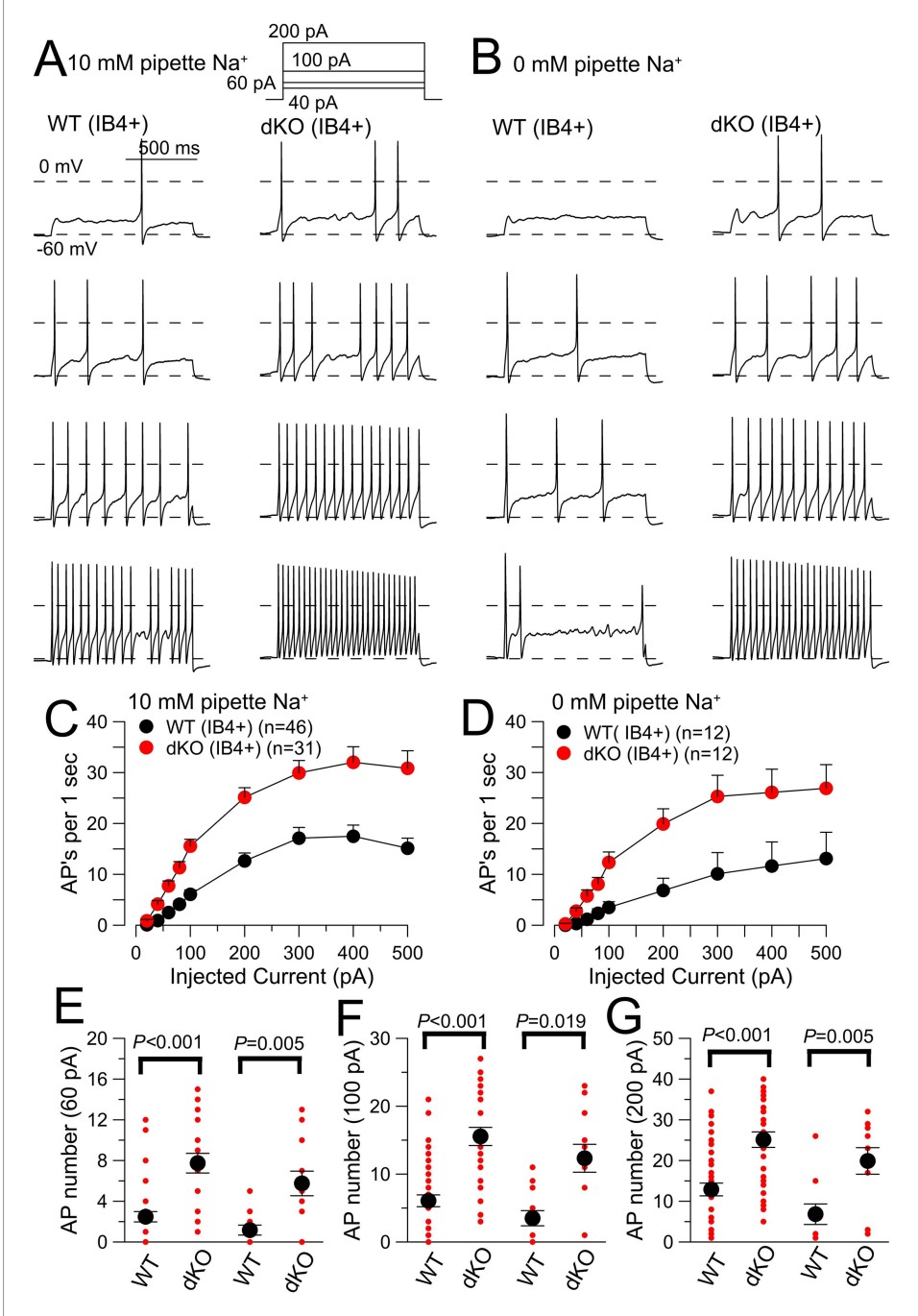

**Figure 7**. Evoked action potential (AP) firing is increased in IB4+ DRG neurons from Slo2 dKO mice. (**A**) 40, 60, 100, and 200 pA current injections (1 s) were used to elicit firing in WT (left) and dKO IB4+ DRG neurons from a holding potential of −60 mV. The pipette solution contained 10 mM Na+. (**B**) Similar injected currents were used to elicit firing in WT and Slo2 dKO neurons, but with 0 mM pipette Na+. (**C**) Mean number of APs for each 1 s step is plotted as a function of injected current amplitude for WT and dKO neurons for 10 mM pipette Na+. WT and dKO AP firing was significantly different at all injected current levels. (**D**) Mean firing is compared for WT and dKO neurons recorded with 0 mM pipette Na+. (**E**) Mean (black circle) and individual estimates (red circles) of AP firing for 1 s 60 pA current injections are summarized for 10 and 0 mM pipette Na+. p values, KS statistic. For comparisons between 0 and 10 mM Na+, for WT cells, p = 0.909; for dKO cells, p = 0.545. (**F**) AP firing for 100 pA current injections. Between 0 and 10 mM Na+, for WT cells, p = 0.585; for dKO cells, p = 0.245. (**G**) AP firing for 200 pA current injections. Between 0 and 10 mM Na+, for WT cells, p = 0.09; for dKO cells, p = 0.23.

**Table 1**. Properties of IB4+ WT and Slo2 dKO DRG neurons (10 and 0 mM pipette $Na^+$)

| Pipette $Na^+$ | IB4+ WT | | | IB4+ dKO | | | p-values |
|---|---|---|---|---|---|---|---|
| **10 mM $Na^+$** | mean | sem | n | mean | sem | n | **K-S statistic** |
| Cm (pF) | 16.1 | 0.3 | 64 | 15.9 | 0.5 | 41 | 0.574 |
| m.p. (mV) | −54.2 | 0.6 | 57 | −50.8 | 0.9 | 41 | **0.001** |
| $R_{in}$ (MΩ) | 1251.1 | 130.8 | 13 | 1212.9 | 148.8 | 13 | 0.828 |
| rheobase (pA) | 86.6 | 4.6 | 44 | 58.1 | 3.4 | 31 | **0.000** |
| dV/dt AP threshold (mV) | −25.31 | 0.64 | 14 | −27.89 | 0.65 | 10 | 0.032 |
| AP peak (mV) | 39.2 | 2.2 | 14 | 41.7 | 1.6 | 10 | 0.877 |
| AP half-width (ms) | 5.7 | 0.3 | 14 | 5.6 | 0.3 | 10 | 0.771 |
| AHP (mV) | −74.0 | 0.4 | 14 | −72.6 | 0.5 | 10 | 0.124 |
| **60 pA AP count** | **2.3** | **0.5** | **64** | **9.7** | **1.9** | **41** | **0.000** |
| **100 pA AP count** | **5.5** | **0.8** | **64** | **17.7** | **2.7** | **41** | **0.000** |
| **200 pA AP count** | **11.4** | **1.5** | **64** | **28.9** | **5.1** | **41** | **0.000** |
| Pipette $Na^+$ | IB4− WT | | | IB4− dKO | | | p-values |
| **0 $Na^+$** | mean | sem | n | mean | sem | n | **K-S statistic** |
| Cm (pF) | 16.7 | 0.9 | 12 | 16.7 | 0.8 | 12 | 0.991 |
| **m.p. (mV)** | **−54.0** | **1.4** | **11** | **−47.0** | **1.4** | **12** | **0.007** |
| $R_{in}$ (MΩ) | 1381.0 | 194.5 | 12 | 1136.4 | 95.0 | 11 | 0.459 |
| **rheobase (pA)** | **92.5** | **8.7** | **12** | **60.8** | **5.1** | **12** | **0.0048** |
| **dV/dt AP threshold (mV)** | **−22.7** | **0.7** | **10** | **−25.8** | **0.5** | **10** | **0.001** |
| AP peak (mV) | 45.3 | 2.9 | 10 | 51.7 | 1.6 | 10 | 0.313 |
| AP half-width (ms) | 5.1 | 0.3 | 10 | 4.5 | 0.2 | 10 | 0.313 |
| AHP (mV) | −72.5 | 0.7 | 10 | −73.7 | 0.4 | 10 | 0.313 |
| **60 pA AP count** | **1.2** | **0.5** | **12** | **5.8** | **1.2** | **12** | **0.005** |
| **100 pA AP count** | **3.5** | **1.1** | **12** | **12.3** | **2.1** | **12** | **0.019** |
| **200 pA AP count** | **6.8** | **2.4** | **11** | **19.9** | **3.0** | **10** | **0.005** |

Cm, cell capacitance; m.p., resting potential; $R_{in}$, input resistance measured by current deflection arising from a 10 mV pulse from −60 to −70 mV; AP half-width, measured at half peak amplitude; AHP, measured following a single evoked AP; AP count, number of APs in 1 s of specified injected current. Rheobase, defined as smallest injected current which elicited an action potential during a 20 ms current injection.
AP, action potential.

the resting conductance measured from a fit of the I–Vs between −120 and −60 mV, suggesting that there is little obvious basal activation of $K_{Na}$ current with 0 mM pipette $Na^+$. Although these results also indicate that, in the voltage range of −50 to −20 mV, there are other $Cs^+$-sensitive $K^+$ conductances besides $K_{Na}$ active at the end of 20 ms steps, these do not appear to differ significantly between WT and dKO neurons, again supporting the idea that the observed differences in excitability are likely to arise from changes in $K_{Na}$ alone.

Given that in many other cells $K_{Na}$ may play a role in slow AHPs, we also specifically addressed this question in WT DRG neurons. For example, in cells of the thalamic paraventricular neurons (*Zhang et al., 2010a*), it has been shown that trains of APs produce a slow development of $Na^+$-dependent AHPs dependent on the number and frequency of APs in the trains. We therefore examined the consequences of an increasing number of APs on AHPs in IB4+ small diameter neurons. Trains of 5 or 10 APs applied at 7 Hz were unable to elicit any slow AHP in IB4+ small diameter neurons different from that elicited by a single AP (*Figure 10*). This further suggests that $K_{Na}$ current in DRG neurons, at least with physiological ionic solutions, contributes negligibly to membrane potential regulation following APs.

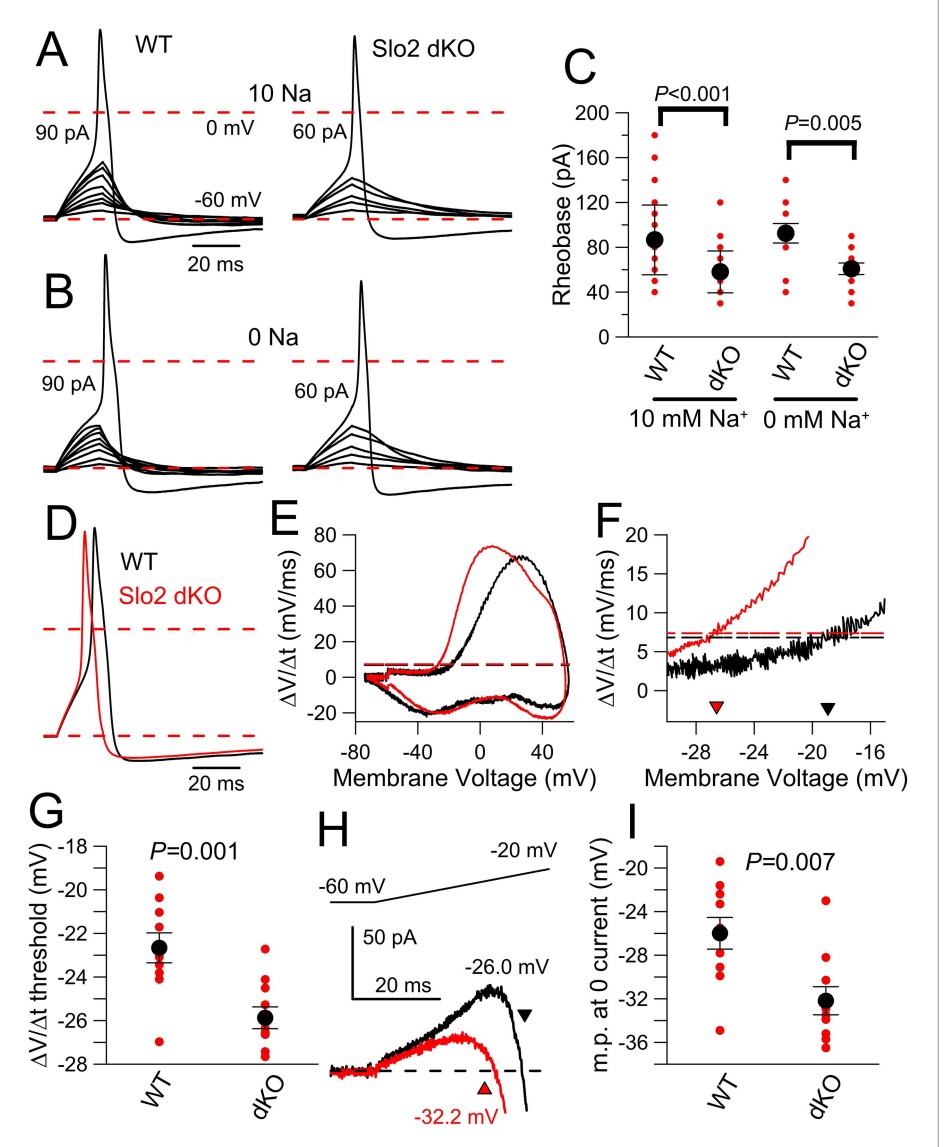

**Figure 8**. Slo2 dKO results in reduced AP threshold. (**A**) A 20 ms current injection of different amplitudes applied with membrane potential adjusted to −60 mV was used to examine AP threshold for a WT (left) and a Slo2 dKO (right) DRG neuron with 10 mM pipette Na$^+$. Dotted red lines indicate 0 and −60 mV voltage levels. Current injection amount that first elicited an AP is indicated on each panel. (**B**) A similar comparison of AP threshold for a WT (left) and dKO (right) neuron is shown with 0 mM pipette Na$^+$. (**C**) Mean and individual determinations of effective rheobase as determined in panels (**A**) and (**B**) are plotted for WT and dKO cells both for 10 and 0 mM pipette Na$^+$. p values are the KS statistic for the indicated pairs. There was no difference for comparisons of 0 and 10 mM Na$^+$ within a given genotype. (**D**) Example single APs elicited by a 100 pA current injection for WT and Slo2 dKO neurons are shown (0 mM pipette Na$^+$). (**E**) dV/dt is plotted as function of membrane voltage for the APs in panel (**D**) (dKO, red; WT, black). Horizontal dotted lines correspond to the dV/dt value that is 10% of peak dV/dt for a given cell. (**F**) The dV/dt plot is shown for a more limited range of membrane voltage, with crossover with horizontal dotted lines of same color showing effective AP threshold. (**G**) Thresholds determined from dV/dt analysis are plotted for WT and dKO neurons (p = 0.001, KS statistic). (**H**) Currents activated by a 40 ms voltage-ramp from −60 to −20 mV from a holding potential of −60 mV were averaged for 10 WT and 11 dKO neurons (0 mM pipette Na$^+$). The voltage at which the current becomes net inward is indicated by the arrows. (**I**) The membrane potential at which net current becomes inward during the voltage-ramp protocol shown in (**H**) is plotted for WT and dKO neurons with p = 0.007 (KS statistic).

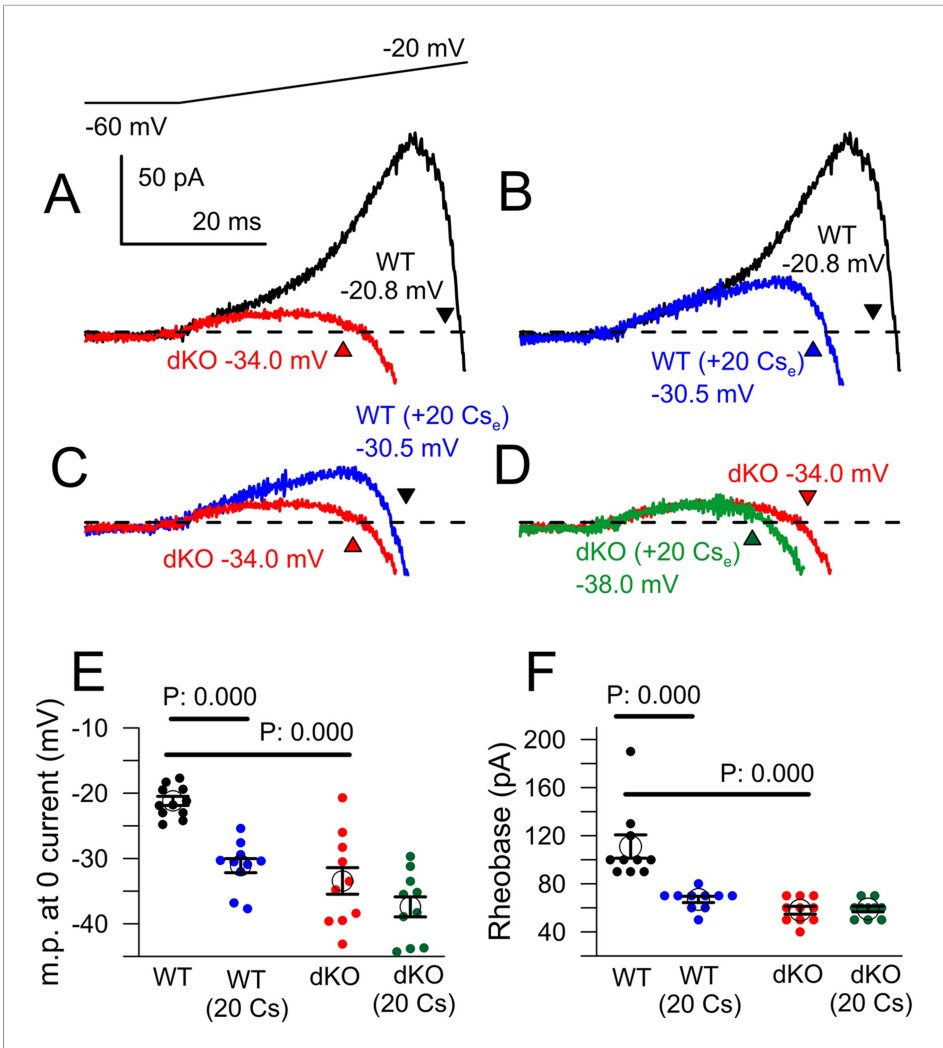

**Figure 9**. Cs⁺ inhibition of outward current in WT, but not dKO, neurons recapitulates properties of Slo2 dKO. (**A**) Traces show averaged currents activated by the indicated voltage-ramp protocol (top) for 10 WT and 10 dKO neurons. Number shows the voltage at which net current crosses the 0-current level (indicated approximately by arrow heads). (**B**) Ramp-activated currents are shown for WT cells before and after application of 20 mM extracellular Cs⁺. (**C**) Ramp-activated currents are compared for WT cells in the presence of 20 mM Cs⁺ and dKO cells. (**D**) Currents are shown for dKO neurons without and with 20 mM extracellular Cs⁺. (**E**) The mean 0-current potential for sets of WT and dKO neurons without and with Cs⁺ are plotted, along with the individual estimates from each cell. p values are KS statistics. Other comparisons had p-value estimates >0.1. (**F**) Mean rheobase for the same set of WT and dKO cells are plotted, along with individual estimates.

The following figure supplements are available for figure 9:

**Figure supplement 1**. Normalized ramp-activated currents reveals that application of Cs⁺ shifts the apparent range of inward current activation in a fashion similar to dKO of Slo2 currents.

**Figure supplement 2**. Comparison of step-activated inward and steady-state currents activated in WT and dKO cells for comparison of ramp-activated outward current.

**Figure supplement 3**. Evaluating the potential impact of a small K⁺ conductance near resting potential.

Overall, these results suggest that the contribution of $K_{Na}$ to DRG firing behavior is relatively insensitive to resting cytosolic Na⁺ levels up to 10 mM. The absence of a difference in resting conductance between WT and dKO neurons at potentials between −60 and −120 mV suggests that

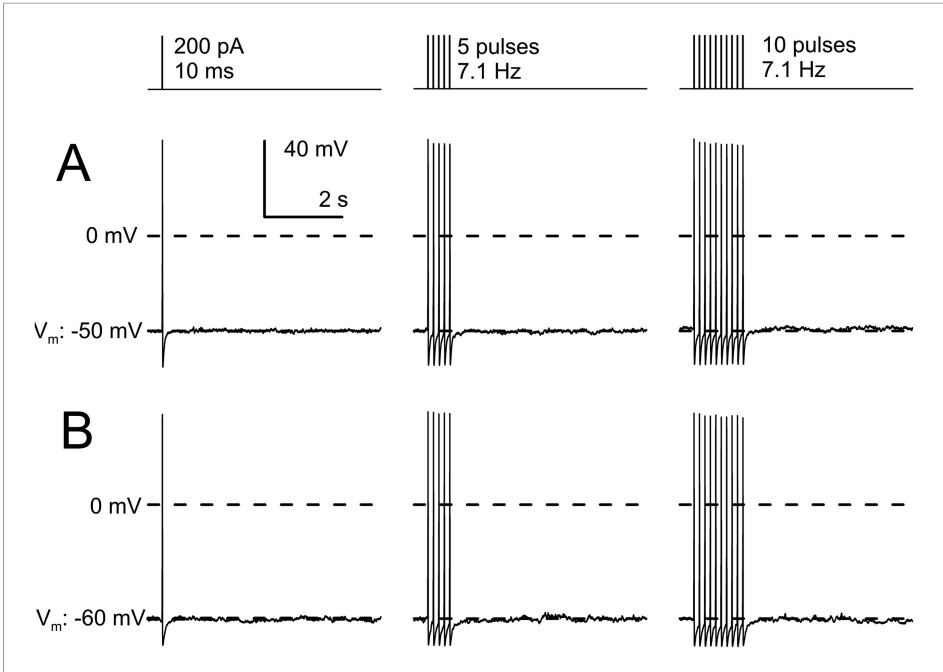

**Figure 10**. AP trains in IB4+ small diameter DRG neurons do not develop slow AHPs. (**A**) A cell was maintained at a resting potential of −50 mV and stimulated with either 1, 5, of 10 pulses of 10 ms duration and 200 pA amplitude, with a pulse frequency of 7.1 Hz. A brief afterhyperpolarization is associated with the last AP in each test, with no indication of any additional slow afterhyperpolarization persisting for 100 s of milliseconds. (**B**) In another cell maintained at a resting potential of −60 mV, the identical current injection sequence also failed to elicit any slow afterhyperpolarization. Identical results were observed in three additional neurons.

basal $K_{Na}$ activation is minimal. Therefore, an influence of $K_{Na}$ current at the foot of AP generation presumably requires a source of $Na^+$ and perhaps a requirement for coupling to local $Na^+$ influx. Although it has been proposed that $K_{Na}$ channels may be coupled to influx through particular subtypes of Nav channels (*Hage and Salkoff, 2012*), the results here would require that such coupling must be very tight and occur immediately upon Nav activation. Future work will be required to address these issues, but the results here suggest new considerations on the conditions under which $K_{Na}$ activation may occur.

## Discussion

Direct evidence supporting the existence of $K_{Na}$ channels in native cells first appeared about 25–30 years ago (*Kameyama et al., 1984*; *Bader et al., 1985*; *Dryer et al., 1989*). Yet, even with identification of the two mammalian Slo2 genes that encode $K_{Na}$ channels of the type observed in the early studies (*Bhattacharjee et al., 2003*; *Yuan et al., 2003*), full definition of functional properties and physiological roles of $K_{Na}$ currents have proven somewhat elusive, despite the apparently widespread distribution of Slo2 subunits in excitable tissue (*Bhattacharjee et al., 2002*, *2005*). Recent demonstration of neurological disorders linked to Slo2.2 (*Barcia et al., 2012*; *Martin et al., 2014*) further highlights the potential importance of such channels. The availability of Slo2.1 and Slo2.2 KO mice will now provide an additional tool to probe the roles of $K_{Na}$ currents. The present results demonstrating the loci of expression of message for Slo2.1 and Slo2.2 subunits, the presence of Slo2.1 and Slo2.2 protein in various tissues, and the natural occurrence of Slo2.1/Slo2.2 heteromultimers confirms and extends earlier work regarding the loci of expression of Slo2 subunits (*Bhattacharjee et al., 2002*, *2003*; *Chen et al., 2009*; *Tamsett et al., 2009*), while also providing KO controls confirming the utility of antibodies in western blots. Here, focusing on the role of Slo2.2 in sensory function, we observed that Slo2.2, but not Slo2.1, KO results in enhanced itch and, to a lesser extent, pain responses. Furthermore, KO of Slo2.2, but not Slo2.1, results in complete absence of the

DRG $K_{Na}$ current. This loss of Slo2 current results in increased excitability in response to depolarizing stimuli, likely accounting for the observed phenotypes.

An important point of the present results is that the primary effect of $K_{Na}$ removal is to reduce AP threshold, with little or no clear effect on AHPs following an AP. The effect on AP threshold was revealed in multiple kinds of tests, a decrease in rheobase, a negative shift in AP threshold determined from the rate of AP rise (dV/dt), and also from a similar negative shift in the voltage at which current becomes net inward during a ramp protocol. Similar changes in AP threshold were observed with both 0 and 10 mM pipette $Na^+$, excluding a key role for differences in pipette $Na^+$ as a determinant of basal $K_{Na}$ activation. Overall, the results require that $K_{Na}$ acts as a mild brake to the onset of AP initiation. Furthermore, we observed no difference in AHP amplitude measured following single APs between WT and KO cells, either with 0 or 10 mM pipette $Na^+$. Trains of APs also did not evoke the development of slower AHPs.

A role of $K_{Na}$ activity preceding the AP upswing in DRG neurons differs fundamentally from other explanations of the proposed role of $K_{Na}$ current in other cells. In different cases, a primary role for $K_{Na}$ current has been proposed either in fast (*Hess et al., 2007*; *Yang et al., 2007*; *Gribkoff and Kaczmarek, 2009*; *Markham et al., 2013*) or slower afterhyperpolarizations that may require trains of several APs to develop (*Schwindt et al., 1989*; *Kim and McCormick, 1998*; *Franceschetti et al., 2003*; *Wallen et al., 2007*; *Zhang et al., 2010a*). In the case of rapid coupling of $K_{Na}$ activation to single APs, this potentially provides a mechanism to facilitate high frequency firing (*Yang et al., 2007*), while slow AHP development serves to support AP accommodation and termination of burst activity. The present results do not preclude such roles for $K_{Na}$ either in rapid repolarization or slow AHPs in other cells. However, the specific role of $K_{Na}$ channels in any cell would depend intimately on the balance of other repolarizing conductances, along with magnitude, spatial, and temporal properties of any cytosolic $[Na^+]$ elevation. In this regards, it is worth mentioning that some of the best support for the presence of $K_{Na}$ currents in cortical neurons has required conditions under which $Ca^{2+}$-dependent outward currents are inhibited (*Schwindt et al., 1989*). Given the abundance of other repolarizing conductances in DRG neurons, it is perhaps not unexpected that modest $K_{Na}$ activation during AP repolarization in DRG neurons may have negligible effects on $V_m$. Our observations are, in fact, consistent with earlier results in rat DRG neurons, in which both AP duration and AHPs were unchanged when extracellular $Na^+$ was replaced by $Li^+$ (*Bischoff et al., 1998*), despite the fact that $Li^+$ does not substitute for $Na^+$ in $K_{Na}$ activation (*Safronov and Vogel, 1996*).

If $K_{Na}$ activity acts as a brake to AP initiation, how is $K_{Na}$ activation elicited? Despite the well-established existence of $K_{Na}$ channels, the circumstances under which cytosolic $Na^+$ elevation arising from physiological stimuli is sufficient to produce $K_{Na}$ activation remain unclear. In fact, consideration of basic properties of $Na^+$ diffusion and the expected $Na^+$ flux through single channels have raised some doubt whether average $[Na^+]_i$ can ever be sufficient to activate $K_{Na}$ (*Dryer, 1991*). Some aspects of our data partially address these issues, but there are complexities in our observations that are not readily explained. The differences in firing properties of WT and dKO cells, both with 0 and 10 mM pipette $Na^+$, suggest that $K_{Na}$ activation is unaffected over the range of 0–10 mM pipette $Na^+$. This is not surprising given that the threshold for $K_{Na}$ activation may be higher than 10 mM (*Bischoff et al., 1998*; *Tamsett et al., 2009*). Furthermore, when $R_{in}$ was measured with a step from −60 to −70 mV, no difference between WT and dKO neurons was observed either with 0 and 10 mM pipette $Na^+$. Similarly, with $R_{in}$ measured by a fit to the I–V relationship over voltages from −120 to −60 mV with 0 $Na^+$ pipette solution, no $Cs^+$ dependent inhibition of conductance was observed in either WT or dKO neurons. Although it has been suggested that some Slo2.2 channel activation may occur in 0 $Na^+$ (*Huang et al., 2013*), the present results suggest that in DRG neurons basal $K_{Na}$ activity at potentials negative to −60 mV does not occur. However, both with 0 and 10 mM pipette $Na^+$, WT cells exhibited a slightly more negative $V_m$ than dKO cells. If pipette $Na^+$ itself does not influence the differences in WT cells from dKO cells, how might these differences arise? The largely linear behavior of both the WT and dKO neuron steady-state I–V relationship from −120 mV to −60 mV begins to exhibit distinct upward curvature in the range of −60 to −50 mV, bracketing the range of measured membrane potentials. This would suggest that conductances are active at rest that are apparently not active negative to −60 mV. Based on the differences in WT and dKO resting potentials, $K_{Na}$ current is clearly a candidate for one of these conductances. In addition, that $V_m$ is close to −50 mV with $E_K \sim$ −80 mV suggests that there may be appreciable inward Na current at potentials above −60 mV. Future work will be required to assess the identity of any components of $Na^+$ current active at such potentials.

Voltage-step protocols used here from a holding potential of −70 mV reveal little obvious inward current until at least −30 mV. Both Nav1.8 and Nav1.9 channels are known to be expressed in some small diameter IB4+ neurons (*Fang et al., 2006*; *Strickland et al., 2008*) and Nav1.9, in particular, may begin to activate at potentials close to resting potentials we have observed (*Rugiero et al., 2003*; *Coste et al., 2004*; *Zhao et al., 2011b*). Another possibility reflects the proposal that $K_{Na}$ currents in some neurons may be selectively activated by persistent TTX-sensitive $Na^+$ currents (*Hage and Salkoff, 2012*). TTX-sensitive Nav1.7 channels can be found in small diameter DRG neurons (*Nassar et al., 2004*) and, although such channels are largely inactivated near DRG resting potentials (*Vijayaragavan et al., 2001*), it is possible that even under steady-state inactivated conditions some persistent openings occur. Perhaps as Slo1 $Ca^{2+}$-dependent $K^+$ channels are coupled to specific $Ca^{2+}$ channels (*Berkefeld et al., 2006*; *Berkefeld and Fakler, 2008*), molecular coupling of $K_{Na}$ channels to specific sources of $Na^+$ influx may occur.

Although the changes in rheobase, AP threshold, and increase in excitability observed in the dKO animals can largely consistent with what one would expect from the simple demonstrated removal of $K_{Na}$ current, there is also the possibility that genetic deletion of Slo2 protein may result in compensatory changes that account for some of the observed effects. This issue might be of particular concern in regards to DRG neurons, since it is well-known that a variety of manipulations can readily induce changes in various DRG current properties, including Nav channels, resulting in altered excitability (*Chahine and O'Leary, 2014*). Furthermore, in the particular case of Slo2.2, it has been proposed that severe human pathologies associated with Slo2.2 mutation arise from extensive alterations in gene and protein expression throughout the nervous system (*Kaczmarek, 2013*). In the present case, two possible alternative mechanisms by which rheobase, AP threshold, and excitability might be altered as observed in the Slo2 dKO neurons would be, first, a shift in Nav current activation to more negative potentials and, second, a loss of some other $K^+$ current active in the subthreshold range of voltages. Although we cannot fully exclude that there have been no changes other than loss of $K_{Na}$ current in the Slo2 dKO DRG neurons, the demonstration that inhibition of subthreshold $K^+$ current by $Cs^+$ mimicked the behavior of the dKO neurons strongly argues that a change in Nav channel expression does not underlie the observed phenotypes and, furthermore, that removal of a sub-threshold $K^+$ conductance can produce the particular constellation of changes we have observed. Finally, we did not observe any indication of a loss in a near threshold $K^+$ conductance other than $K_{Na}$, although any such change might be difficult to resolve.

It is instructive to consider how much $K_{Na}$ current activation might be required to account for changes in resting potential. Assuming a simple modified GHK conductance equation and a relative balance of $G_K$, $G_{Na}$, and $G_{Cl}$ (net $R_{in}$ = 858 MΩ; $G_{in}$ = 1.165 nS) to yield a resting potential near −50 mV, increasing the background $G_K$ of 0.7 nS with an additional activation of 0.16 nS $G_{KNa}$ results in additional hyperpolarization of ∼3.3 mV (*Figure 9*, *Figure 9—figure supplement 3*). From *Figure 6F*, we observed an average $K_{Na}$ conductance of 69 nS activated by 70 mM pipette $Na^+$ around −50 mV. Although 70 mM $Na^+$ produces a less than maximal activation, if one assumes a maximal conductance of 69 nS, the fractional activation of $K_{Na}$ required to produce a 3–4 mV hyperpolarization corresponds to a Po of about 0.002, which corresponds to 8 pA at −50 mV. If these calculations are generally correct, it is not surprising that it would be difficult to identify procedures to directly examine such a current.

The severity of the human patients with apparent Slo2.2 mutations (*Barcia et al., 2012*; *Heron et al., 2012*) naturally raises a question regarding whether suitable phenotype tests may uncover cognitive impairments in the Slo2 dKO mice. Any such deficits, if they exist, apparently spare the basic ability of the dKO animals to eat, mate, and function in a generally normal way. Perhaps relevant to the possibility that Slo2.2 KO may have apparently benign functional consequences, one set of the human Slo2.2 mutations corresponds to gain-of-function changes (*Barcia et al., 2012*), resulting in larger $K_{Na}$ currents. Perhaps the presence of Slo2.2 subunits of altered function results in more deleterious consequences than the complete absence of such subunits.

Recent work on another Slo2.2 KO model (*Lu et al., 2015*) also focused on sensory function with some complementary results. In both cases, exon 11 of the gene encoding Slo2.2 was deleted. Both groups observe similar absence of effects of Slo2.2 KO on hotplate and formalin tests. Given the absence of effects on acute pain responses, *Lu et al. (2015)* focused on neuropathic pain responses, observing that Slo2.2 activation reduces neuropathic pain and does not acutely influence sensory responses. However, our results clearly show that Slo2.2 KO influences the immediate response to

sensory stimuli, in particular, itch. Furthermore, an enhancement of the acute responses to capsaicin also occurs at lower doses. Both groups also observed increased excitability in neurons lacking $K_{Na}$ current, although the basis for the enhanced excitability was not examined in detail by the other group (*Lu et al., 2015*). However, we suggest that the increase in DRG neuron excitability observed in both studies is consistent with enhancement of the immediate response to a sensory stimulus.

Why do some aversive tests, for example, hotplate, formalin, tail flick, and cold plate (*Lu et al., 2015*), show no difference between WT and Slo2.2 KO? We envision three possible explanations. First, as suggested by the dose-dependence of capsaicin responses, perhaps regulation of $K_{Na}$ current has more impact on weaker stimuli or relatively weak depolarizing drive, whereas, with stimuli that elicit strong initial depolarization, modest $K_{Na}$ activation will be less likely to influence AP generation. Second, even if $K_{Na}$ is present in most small diameter DRG neurons, different categories of such neurons may have other conductances that diminish the impact of loss of $K_{Na}$ current. Third, perhaps there are small diameter neurons, or certainly DRG neurons of other sizes, that may not have $K_{Na}$ current.

In sum, we propose that, in small diameter IB4+ DRG neurons, $K_{Na}$ currents influence AP onset, with greatest effect during low frequency firing. The particular properties of $K_{Na}$ currents, specifically modest intrinsic voltage-dependence but voltage-dependence perhaps acquired through coupling to its cytosolic ligand, $Na^+$, may be well-suited to influence the initial upswing of AP generation, at a time when other $K^+$ conductances are largely quiescent.

## Materials and methods

### Animal care

Animals were handled and housed according to the National Institutes of Health Committee on Laboratory Animal Resources guidelines. All experimental protocols (protocol #20130256) were approved by the Washington University in St Louis Institutional Animal Care and Use Committee. Every effort was made to minimize pain and discomfort.

### Generation of KO mice

To generate the Slo2.1 KO (deletion of *Kcnt2* exon) mouse, exon 22 (110 bp, encoding amino acids 829–865 of the Slo2.1 protein) was targeted for deletion. The deletion of exon 22 in *Kcnt2* causes a frame-shift and the predicted residual protein is Slo2.1(1–828). To generate the Slo2.2 KO (deletion of *Kcnt1* exon) mouse, exon 11 (181 bp, encoding amino acids 253–313 of Slo2.2, which includes part of the pore-forming region) was targeted for deletion. The deletion of exon 11 in *Kcnt1* causes a frame-shift and the predicted residual protein is Slo2.2(1–252) with an appended 26 amino acid peptide before the first stop codon. Following germline transmission via recombinant ES cells, the F1 mice with targeted loci were bred with FLP delete mice (B6.129S4-Gt(ROSA)26Sor[tm1(FLP1)Dym]/RainJ, Jackson Labs, Bar Harbor, ME, United States) to generate floxed mouse lines and with early embryonic expression Cre-mice (EIIa-Cre, Jackson) for deletion of the targeted exons. The *Kcnt1* floxed mice and *Kcnt2* floxed mice are available at The Jackson Laboratory as Stock No. 028418 and Stock No. 028419, respectively. Slo2.1 KO and Slo2.2 KO strains of mice have been maintained in a C57BL/6 background out to N = 12. Additional details of the generation of Slo2.1 and Slo2.2 KO mice are provided in the legend to *Figure 1*. All procedures related to animal care and treatment conformed to institutional and NIH guidelines.

### Behavioral testing

Mice were maintained in a 12 hr light/dark cycle with free access to food and water. Behavioral experiments were done on male mice of 10–12 weeks of age and mice were only used once for any of the tests. Littermate mice were used in all behavioral studies, except those involving double KO of Slo2.1 and Slo2.2. For dKO mice, each allele had initially been breed within a C57BL/6 background out to N = 12 generations, so comparisons were made to the Jackson Labs WT C57BL/6 stock. Animals were habituated to the experimental room with background white noise used to mask random noise (SKI 000148, San Diego Instruments, San Diego, CA, United States) and monitored by an observer naïve to genotype.

#### Hot plate test

The PE34 Hot Plate Analgesia meter (IITC Life Science, Woodland Hills, CA, United States) was used for heat latency testing, following published procedures (*Zhao et al., 2011a*). Mice were placed on a

black anodized aluminum plate within a round enclosure (diameter, 10 cm; height, 15 cm). The plate's surface temperature was adjusted to 55°C (±0.1°C) and the temperature was constant throughout the tests. The time between placement on the plate and a defensive movement (hindpaw licking or jumping) was recorded. Cut off time was set at 20 s.

### Formalin test
Formalin (15 µl of a 0.5% formaldehyde solution) was injected subcutaneously into the dorsal surface of one hindpaw (*McNamara et al., 2007*). The time spent licking the formalin-injected paw was recorded in 5 min intervals up to 45 min after formalin injection.

### Itch tests
Mice were shaved at the back of the neck. Following intradermal injections of potential pruritic agents (EtOH, CQ, HA, compound 48–80), hindlimb scratching behavior directed towards the shaved area was monitored over a 30 min period. CQ and compound 48–80 were dissolved in 0.35% EtOH. HA was prepared in a saline solution.

### Capsaicin injections
Mice were placed into a transparent observation chamber (30 × 30 × 25 cm) for adaptation 30 min before the experiments (*Kim et al., 2001*). Capsacin was administered subcutaneously into the plantar hindpaw in a volume of 10 µl using a 50 µl Hamilton syringe attached to a 30 gauge needle. The needle was inserted at the midline near the heel and advanced anteriorly to the base of the second or third toe, where the drug was injected, forming a bleb that usually extended back to the initial point of entry. Capsaicin (Sigma Aldrich, St. Louis, MO, United States) was first dissolved in 95% ethanol (100 µg/µl) before diluting to the desired concentrations in PBS. After injections, mice were then placed into the original chamber and were observed for licking and flinching behavior. Time spent licking or flinching was recorded in 5-min sections, for a total of 15 min.

### Cheek test
The hair on a patch of each cheek was shaved at least 2 days prior to experiments (*Shimada and LaMotte, 2008*). While under mild restraint, the cheek was then injected, all within 10 s. During the following 30 min, both forepaw wiping motions and hindlimb scratching motions were separately counted and grouped into 5 min bins.

## RNA extraction and quantitative RT-PCR
RNA extraction and RT-PCR followed procedures previously used in this laboratory (*Yang et al., 2011*; *Martinez-Espinosa et al., 2014*). Total RNA from different mouse tissues was isolated using the RNeasy Plus Mini Kit (Qiagen, Valencia, CA, United States) following the manufacturer's recommendations. Before the reverse transcription, total RNA was treated to remove genome DNA with the DNA-free Kit (AM1906, Applied Biosystems, Waltham, MA, United States). cDNA was synthesized using the Retroscript Kit (AM1710, Applied Biosystems). For the negative control groups, all components except the reverse transcriptase MMLV-RT were included in the reaction mixtures. Real-Time PCR with specific primers (*Table 2*) was performed using Power SYBR Green PCR Master Mix (Applied Biosystems). Mouse β-actin gene was utilized here as the homogenous standard. The running protocol extended to 40 cycles consisting of 95°C for 15 s and 60°C for 1 min using an Applied Biosystems 7500 Fast Real-time PCR system. PCR specificity was checked by dissociation curve analysis and DNA electrophoresis. Primer efficiency was validated as previously reported (*Yang et al., 2009*). Abundance was calculated from $2^{-dCt}$, with $dCt = Ct(target) - Ct(\beta\text{-}actin)$. Each reported estimate is the average from three separately prepared mouse tissue RNA samples, with each sample run in triplicate.

## Mouse tissue total protein and membrane protein preparations
Preparation and analysis of proteins from mouse tissues followed procedures recently used in this laboratory (*Yang et al., 2011*; *Martinez-Espinosa et al., 2014*). Mature male mice were sacrificed for preparation of membrane proteins from whole brain, cerebellum, cortex and spinal cord, respectively. 1 g of mouse whole brain, cortex, cerebellum, or spinal was homogenized with Teflon-glass pestle in 10 ml ice-cold 0.32 sucrose in PBS, including 100 µl 1.5 M PMSF in acetone and 100 µl Protease Inhibitor Cocktail (Sigma-Aldrich). After spinning at 300×g for 10 min at 4°C, the supernatant was collected, followed by ultra-speed centrifugation in a 4°C Ti70 rotor at 150,000×g for 1 hr. The

**Table 2.** Primers used for Real-Time PCR

| Gene | Primer | Amplicon length |
|---|---|---|
| Kcnt2 | Forward: 5'-TCTATTTGAAACAATACTCCTTGG-3' | 149 bp |
| | Reverse: 5'-GAACAAATAGATTTCTTAAGGTGG-3' | |
| Kcnt1 | Forward: 5'-CTCACACACCCTTCCAACATGCGG-3' | 161 bp |
| | Reverse: 5'-ATGCTGATACTAAATACTCGACCA-3' | |
| B-actin | Forward: 5'-TGGAGAAGAGCTATGAGCTGCCTG-3' | 127 bp |
| | Reverse: 5'-GTAGTTTCATGGATGCCACAGGAT-3' | |

membrane pellet was resuspended in 10 ml lysis buffer (50 mM Na phosphate, 150 mM NaCl, 10 mM KCl, 2% Triton X-100, pH 7.2), including 100 µl 1.5 M PMSF in acetone and 100 µl Protease Inhibitor Cocktail, and rocked at 4°C for 1 hr, followed by centrifugation at 14,000×g for 10 min. 10 ml supernatant was saved as the membrane protein preparation in the −80°C freezer. Hearts from four mature male mice were dissected, washed with PBS and quickly frozen in liquid nitrogen. The frozen hearts were pulverized with liquid nitrogen pulverizer and then homogenized on ice with Teflon-glass pestle in 3 ml TE(pH 7.6) buffer containing 2% Triton X-100, 20 µl PMSF (1.5 M in acetone) and 20 µl Protease Inhibitor Cocktail. The suspension was rocked at 4°C cold room for 1 hr, followed by spinning at 14,000Xg for 15 min. Pellet was discarded and the 3 ml supernatant was saved as the heart total protein preparation. DRGs collected from 10 mature male mice were homogenized on ice with a Teflon-glass pestle in 1 ml lysis buffer (50 mM Na phosphate, 150 mM NaCl, 10 mM KCl, 2% Triton X-100, pH 7.2), including 10 µl 1.5 M PMSF in acetone and 10 µl Protease Inhibitor Cocktail. The suspension was rocked at 4°C cold room for 1 hr, followed by spinning at 14,000×g rpm for 15 min. The 1 ml supernatant was saved as the DRG total protein preparation.

## Immunoprecipitation and western blotting

Samples of total protein preparations or membrane protein preparations appropriate for a given tissue were applied in the immunoprecipitation experiment. 70 µl Protein A/G Plus agarose beads (Santa Cruz Biotechnology, Dallas, TX, United States) were added to the preparation and mixed at 4°C cold room for 1 hr. The beads were removed by a brief spin at 14,000×g. The supernatant was carefully collected and mixed with 8 µg monoclonal anti-mSlo2.1 (N11/33) or anti-mSlo2.2 (N3/26) antibody (Antibodies Inc., Davis, CA, United States) at 4°C cold room for 2 hr, followed by the addition of 80 µl Protein A/G Plus agarose beads. The mixture was rocked overnight and then centrifuged briefly to collect the beads. The beads were washed three times with 1 ml 1% Triton X-100 in PBS and the bound proteins were eluted from the beads with 100 µl SDS loading buffer containing 100 mM DTT.

For western blotting, aliquots of total protein preparations or membrane protein preparations was mixed well with an equal volume of 2× SDS loading buffer containing 100 mM DTT, maintained at room temperature for 30 min before loading onto 8% Precise Protein Gels (Pierce, Life Technologies, Grand Island, NY, United States). Protein markers were EZ-Run Prestained Rec Protein Ladder (Fisher, Waltham, MA, United States). Proteins were transferred to Immobilon-P PVDF membranes with the Trans-Blot Semi-Dry Transfer System (Bio-Rad, Hercules, CA, United States). Membranes were blocked with 5% nonfat milk in Tris-buffered saline-Tween 20 solution (pH 7.3) at room temperature for 1 hr, followed by overnight incubation at 4°C in 5 ml blocking solution containing monoclonal anti-mSlo2.1 or anti-mSlo2.2 antibody (10 µg/ml, Antibodies Inc). After washing with 5 ml blocking solution × 5 min for four times, membranes were incubated with 5 ml blocking solution containing Mouse Trueblot Ultra HRP-conjugated anti-mouse IgG (at 1:2000 dilution, eBioscience, San Diego,

CA, United States) at room temperature for 1 hr. After four-time washing with 5 ml blocking solution, HRP-labeling was developed using Amersham ECL Plus Western Blotting Detection System (GE Healthcare, Pittsburgh, PA, United States). A specific Slo2.1 band in WT heart Slo2.1-IP sample was not detected in the first round of western blot with the monoclonal anti-Slo2.1 antibody (10 μg/ml, Antibodies Inc.). To visualize a Slo2.1-specific band, the initial western blot was stripped with Re-Blot Plus Mild Solution (Millipore; Billerica, MA, United States) and then the PVDF membrane was reblotted with the same antibody.

For western blotting, NeuroMab anti-Slo2.1 antibody (#75-055) targets amino acids 564–624 of Slo2.1; NeuroMab anti-Slo2.2 antibody (#75-051) is against amino acids 1168–1237 of Slo2.2. Slo2.1 KO predicts a residual protein Slo2.1(1–828) with a predicted MW of 91 kDa, which should be recognized by the NeuroMab antibody, but is not observed in the Slo2.1 KO brain (*Figure 2*). This indicates that, following deletion of exon 22, no residual Slo2.1 protein remains in the knockout mouse, probably due to the instability of the truncated mRNA or protein. In the case of Slo2.2 KO, deletion of exon 11 (encoding amino acids 285–354 of Slo2.2) causes a frame-shift such that the predicted residual Slo2.2(1–252) protein does not contain the sequence recognized by the NeuroMab anti-Slo2.2 antibody. Therefore, no residual Slo2.2 protein was detected in Slo2.2 KO membrane samples. Since the residual Slo2.2(1–252) is terminated in the middle of the S6 segment of the inner helix, it seems unlikely that any residual Slo2.2 protein fragments in the knockout mouse would assemble into functional channels. From *Figure 2A,B,D*, it is clear that knocking out the gene for Slo2.1 has no obvious effect on the presence of Slo2.2 protein, and vice versa.

## Acute DRG dissociation

After removal of DRG from 3 to 5 week old mice, ganglia were desheathed and then incubated in 15 U/ml papain/L-cysteine in HBSS without calcium and magnesium (Life Technologies) for 20 min at 37°C. Ganglia were washed three times in HBSS, replaced with 1.5 mg/ml collagenase (Sigma–Aldrich) in HBSS and incubated for 20 min at 37°C. After washing three times with Neurobasal-A medium supplemented with 10% FBS, B27 supplement, 100 U/ml penicillin/streptomycin, and Glutmax (2 mM L-alanyl-L-glutamine) (all from Life Technologies), ganglia were gently triturated with a flame-polished Pasteur pipette until the solution turned cloudy. The dispersed cells were diluted with growth medium containing supplemented Neurobasal medium. The cells were plated at a density of ~2000 cells per well on 12 mm glass coverslips coated with Matrigel (BD Biosciences, San Jose, CA, United States), and maintained at 37°C in humidified air with 5% $CO_2$ for 1 hr before onset of recording. Most experiments were done within 8 hr after dissociation and changes of the culture medium were not necessary. For 2–3 days in culture, half the medium was replaced with fresh growth medium on the second day.

## DRG slice preparation

100 μm thin slices were prepared from dorsal root ganglia of 7–14 day old of mice using a previously described method (*Safronov et al., 1996*). In brief, mice were killed by $CO_2$ inhalation, rapidly decapitated, and six ganglia from lower thoracic and lumber regions were carefully removed in ice-cold Hank's Balanced Salt Solution (HBSS, Invitrogen; Carlsbad, CA, United States). Ganglia were desheathed using fine forceps, placed in the center of a 35 mm petri dish, then filled with with 40°C 4% low melting agar (wt/vol in HBSS). The dish was then immediately submerged in ice-cold artificial CSF cutting solution, which contained the following (in mM): 125 NaCl, 3.5 KCl, 0.5 $CaCl_2$, 3.5 $MgCl_2$, 26 $NaHCO_3$, and 10 D-glucose. The solution was bubbled with 95%$O_2$/5%$CO_2$ to maintain pH at ~7.4. After solidification of the agar, small blocks containing ganglia were cut out and glued onto the cutting platform of a vibratome (VT100, Leica, Buffalo Grove, IL, United States) for cutting. Slices were stored for 45 min at 35°C and kept at room temperature until recording. The oxygenated storage solution contained the following (in mM): 125 NaCl, 3.5 KCl, 26 $NaHCO_3$, 10 D-glucose, 2.5 $CaCl_2$, and 1.3 $MgCl_2$. Individual slices were subsequently transferred to a recording chamber continuously perfused (3 ml/min) with oxygenated saline at room temperature. A Slicescope Pro 3000 (Scientifica Ltd, East Sussex, United Kingdom) microscope equipped with Nomarski optics, a 40× water-immersion lens, and infrared illumination was used to view DRG neurons in the slices.

## IB4 labeling

Small diameter DRG neurons responsive to itch and pain stimuli (*Stucky and Lewin, 1999*; *Lallemend and Ernfors, 2012*) express a cell surface antigen that binds a plant lectin, isolectin B4 (*Silverman and Kruger, 1990*). To categorize neurons as either IB4+ or IB4−, prior to recording, DRG neurons, whether dissociated or in slices, were exposed to media containing either 5 µg/ml isolectin B4(FITC) or 1 µg/ml isolectin B4(Texas Red). After 5 min incubation, cells were returned to normal extracellular solution and viewed with standard fluorescence microscopy.

## Basic recording methods

Standard whole-cell recording methods were used for both voltage-clamp and current clamp using a Multiclamp Amplifier (Molecular Dynamics, Sunnyvale, CA, United States), for both dissociated cells and for cells in slices. Voltage- and current stimulation protocols and acquisition of voltage and current records were accomplished by Clampex 9.2 (Molecular Dynamics) with analysis of waveforms done via Clampfit. Patch-clamp pipettes typically were of 1.5–2.5 MΩ. Following whole-cell access, cells were used if the series resistance (Rs) was less than 10 GΩ. Rs was compensated 85%. For excised patch experiments, pipettes of similar size were used to form GΩ seals on dissociated DRG neurons before excision. The standard internal solution contained (in mM): 10 NaCl, 135 KCl, 1 MgCl, 5 EGTA, 10 HEPES, 3 Mg-ATP, 0.3 Na-GTP, pH 7.3 adjusted with KOH, OSM ~300. In the nominally zero internal $Na^+$ pipette solution, internal KCl was 145 mM, but contained 0.3 mM Na from Na-GTP. The standard external solution contained the following (in mM): 136.4 NaCl, 5.6 KCl, 2.2 CaCl, 1 $MgCl_2$, 11 D-Glucose, 10 HEPES, pH 7.4 adjusted with NaOH. For inside-out patches, the pipette (external) solution contained (in mM): 5 NaCl, 152.5 KCl, 1 $MgCl_2$, 5 HEPES, pH 7.4 adjusted with KOH; the internal solution contained (in mM) 73.6 KCl, 1 $MgCl_2$, 3 EGTA, 10 HEPES and 70 NaCl (for 70 $Na^+$) or 70 Choline-Cl (for 0 Na), adjusted to pH 7.3 with KOH ($E_K$ = 18.35 mV). Both Tetraethylammonium and tetrodotoxin were added to the external solution at final concentrations of 1 mM and 100 nM, respectively, just before the start of experiments. When $Cs^+$ was used as a non-specific blocker of $K_{Na}$ current (*Bischoff et al., 1998*), $Cs^+$ replaced an equal molar concentration of NaCl.

For recording of 'leak' current, after whole-cell formation, to assess 'leak current', the net difference in current observed from voltage-steps from −80 mV to −120 mV (*Figure 6C,D* and *Figure 6—figure supplement 4*) was monitored (*Bischoff et al., 1998*), either with 0 $Na^+$ in the internal pipette solution to define $Na^+$-independent 'leak' current, or with 70 mM $Na^+$. The 70 and 0 mM sodium pipette solutions contained the following (in mM, with 0 $Na^+$ solutions in parenthesis): 70 (0) NaCl, 73.3 (140) KCl, 1 MgCl, 5 EGTA, 10 HEPES, 3 Mg-ATP, 0.3 Na-GTP, pH 7.3 adjusted with KOH, OsM 290–300. For leak current measurements in slices, the external solution contained the following (in mM): 115 NaCl, 5.6 KCl, 1 $MgCl_2$, 1.8 $CaCl_2$, 11 D-Glucose, 1 $NaH_2PO_4$, 25 $NaHCO_3$, bubbled with 95%$O_2$/5%$CO_2$ to maintain pH at ~7.4. For acutely dissociated DRG, the solution contained (in mM): 136.4 NaCl, 5.6 KCl, 1 $MgCl_2$, 1.8 $CaCl_2$, 11 D-Glucose, 10 Hepes, pH 7.4 adjusted with NaOH solution.

## Statistical analysis

The Kolgoromov–Smirnov test was used to generate the KS statistic, P. For cases in which the number of entries in one or both sample populations was less than 10, a two-tailed, unpaired Student's *t*-test was employed. Data are presented as mean ± sem.

## Acknowledgements

The authors thank the Department of Anesthesiology, Washington Univ. for support of this project. The monoclonal antibodies, anti-Slo2.1 N11/33 and anti-Slo2.2 N3/26, were developed by and obtained from the UC Davis/NIH NeuroMab Facility, supported by NIH grant U24NS050606 and maintained by the Department of Neurobiology, Physiology and Behavior, College of Biological Sciences, University of California, Davis, CA 95616. Slo2.1 and Slo2.2 fusion proteins and *Kcnt1/Kcnt2* cDNAs were provided by Chengtao Yang to NeuroMab for generation of antibodies. We thank R Gereau, M Morales for input during various stages of this work. We thank Dr Qin Liu for comments on the manuscript.

## Additional information

### Funding

| Funder | Grant reference | Author |
|---|---|---|
| National Institute of General Medical Sciences (NIGMS) | GM081748 | Christopher J Lingle |
| National Institute of General Medical Sciences (NIGMS) | GM066215 | Christopher J Lingle |

The funder had no role in study design, data collection and interpretation, or the decision to submit the work for publication.

### Author contributions

PLM-E, Designed and performed experiments, analyzed data, and contributed equivalent roles in completion of this project, Conception and design, Acquisition of data, Analysis and interpretation of data, Drafting or revising the article; JW, Designed and performed experiments, analyzed data, and contributed equivalent roles in completion of this project, Conception and design, Acquisition of data, Analysis and interpretation of data; CY, Conception and design, Acquisition of data, Analysis and interpretation of data, Drafting or revising the article; VG-P, HL, Performed experiments, Acquisition of data, Analysis and interpretation of data; HZ, Performed experiments, Acquisition of data, Contributed unpublished essential data or reagents; X-MX, Designed and performed experiments, analyzed data, and contributed equivalent roles in completion of this project, conceived the project, Conception and design, Acquisition of data, Analysis and interpretation of data, Contributed unpublished essential data or reagents; CJL, Conceived the project, directed the project, designed experiments, analyzed data, and wrote the paper, Conception and design, Analysis and interpretation of data, Drafting or revising the article

### Ethics

Animal experimentation: Animals were handled and housed according to the National Institutes of Health Committee on Laboratory Animal Resources guidelines. All experimental protocols (protocol #20130256) were approved by the Washington University in St Louis Institutional Animal Care and Use Committee. Every effort was made to minimize pain and discomfort.

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
