## [Decision Letter]

Thank you for submitting your work entitled “Knockout of SLO2.2 enhances itch, abolishes K_Na_ current, and increases action potential firing frequency in DRG neurons” for peer review at *eLife*. Your submission has been favorably evaluated by Gary Westbrook (Senior Editor) and three reviewers one of whom, Richard Aldrich, is a member of our Board of Reviewing Editors.

The reviewers have discussed the reviews with one another and the Reviewing Editor has drafted this decision to help you prepare a revised submission:

This manuscript examines the consequences of global knockouts of *slo2.1* and *slo2.2* channels on the electrophysiology of small-diameter DRG neurons and on behavioral tests of acute pain and itch. The authors find that knockout of *slo2.2* channels but not *slo2.1* channels results in somewhat enhanced rapid pain and itch responses, most notably in the scratching induced by chloroquine or histamine injection in the first 5 minutes after injection, when wild-type animals have only small responses. In terms of electrophysiology, the authors do two sets of experiments. In one, they identify a sodium-activated conductance that is induced by dialyzing cells with 70 mM Na and show that this is gone in *slo2.2* KO animals. In the second, they study firing properties of the neurons with either physiological (10 mM) or 0 mM internal sodium. Surprisingly, they find that the *slo2.2*-KO animals have enhanced firing frequencies in either condition, accompanied by a reduction in spike threshold.

The work presented in the paper is done well technically and the results are interesting and novel. Another recent paper ([34], duly cited) also examined *slo2.2* KO mice for both electrophysiology of DRG neurons and for behavioral pain responses (but not itch). Unlike the present manuscript, Lu et al. reported no changes in acute pain responses but only in models of neuropathic pain. There is no real experimental conflict, because in the present manuscript the effects on acute pain were actually quite subtle, and were most clearly illustrated by a change in the dose-response curve for capsaicin injections. In the present manuscript the most dramatic effect in the knockout was a large increase in scratching in the first 5 minutes after injection of chloroquine and histamine, and itching was not tested by Lu et al. Therefore the behavioral effects in this manuscript are novel and interesting.

The electrophysiology in the present manuscript is far more extensive and detailed than in the Lu et al. manuscript, although the basic observation of increased excitability in the *slo2.2*-KO animals is similar.

As explained in detail below, a number of additional experiments need to be included to rule out alternative interpretations of the results. We feel that these can be reasonably accomplished.

There are several major issues in the present paper concerning the interpretation of the electrophysiology experiments. In the first series of experiments, the authors show that a Na-dependent conductance is produced by cell dialysis with 70 mM and that this conductance is lacking in the *slo2.2*-KO animals. In this part of the paper, everything suggests that this conductance is completely lacking in WT cells dialyzed with 0 mM Na. But in the second section of electrophysiology, the authors examine changes in firing behavior in the KO animals and find that the changes in excitability are exactly the same when the cells are dialyzed with 0 mM Na as with 10 mM Na. They then interpret the effects in 0 mM Na as suggesting either that the *slo2.2* conductance can actually be present even in the absence of internal Na, or that there is somehow enough Na-entry even at subthreshold voltages to activate the channels. The first possibility seems completely at odds with the first series of experiments, where they see no difference between WT and KO animals with 0 mM internal Na. This seeming inconsistency is not discussed. And there is no attempt to actually test the second possibility.

The authors never consider another possibility – that the changes in excitability are not directly due to loss of Na-activated K current but rather reflect changes in other conductances occurring as a result of developmental changes occurring in the KO animals, or other effects involving changes in protein synthesis by loss of *slo2.2*, for which there is previous evidence. This seems likely. As a whole, the changes in excitability seem far more consistent with a change in other conductances (e.g. an increase or change in voltage-dependence of voltage-activated sodium current, or possibly a reduction in a voltage-dependent potassium current) than a loss of Na-activated K current.

Specific:

1) As noted, there is a fundamental contradiction between the experiments in Figure 6 showing that the conductance identified as *slo2.2* current is lacking when cells are dialyzed with 0 mM Na and the interpretation of the experiments in Figures 7 and 8 where it is assumed that this conductance is somehow present even with 0 mM internal Na. This discrepancy needs to be addressed. In Figure 6, there is no data for the double KO with 0 mM Na internal. If the conductance with this condition is really the same as WT with 0 Na, then the speculation later that there might be Na-independent activity of the *slo2.2* conductance cannot be true.

2) Also inconsistent with the possibility of a Na-independent conductance from *slo2.2* channels is that the input resistance is not changed between WT and KO animals. (What was the protocol for measuring input resistance? From what voltage? Hyperpolarizxing or depolarizing current steps? How big? All of these are relevant to interpretation.)

3) The changes in excitability seem more consistent with an increase in voltage-dependent sodium current near threshold – possibly a shift in voltage-dependence to more negative voltages – than a change in potassium conductance. In the data in Figure 8, the initial outward current in the KO seems similar to control, but then the inward current seems to activate at more negative voltages. A shift in the voltage-dependent of sodium current is also the most obvious explanation for the changes in in the phase-plane plot in Figure 8. The authors need to test whether there is a change in inward sodium current in the knockout animals by recording with K current blocked.

4) A change in non-voltage dependent potassium conductance might shift the frequency-current curves in Figure 6 to the right, but there is no obvious reason it would result in an increased maximal frequency of firing. This also seems more easily explained by a change in sodium current. Obviously, a change in sodium current would also explain why the effect on excitability are the same with 0 internal and 10 mM internal Na.

5) Another possibility is a loss of a voltage-dependent K current. In the IV curves in Figure 6 shown for the dKO animals with 70 mM internal Na, it is striking that there is no evidence of any voltage-dependent component of K current like that seen for the WT neurons with 0 mM internal Na.

6) More generally, the authors interpret all of their results as if they represent direct effects of a loss of *slo2.2* rather than possible developmental changes. Such changes seem not only possible but likely based on the fact that *slo2.2* mutations in humans seem to have profound developmental effects and with previous evidence that *slo2.2* protein can affect synthesis of other proteins. To quote the final summary of the 2013 review by Kaczmarek, “Human mutations in these channels produce profound effects on neuronal function and development, suggesting that perhaps their biological role may extend beyond simply setting the level of neuronal excitability, and that these channels may influence cytoplasmic biochemical pathways that regulate development, plasticity, and intellectual function.” Obviously, such indirect effects could offer a completely different explanation of the changes in excitability – or even the effects on pain and itch behavior. This needs to be acknowledged and considered in the interpretation in this manuscript.

7) Because this is the first description of these knockouts, the authors should provide some rationale for gene targeting of the particular exons that were selected (perhaps in the Methods?). The authors should address what the selected exons encode, if knockout of the exon conceivably causes a dominant negative protein for its paralogue channel (i.e., is slick exon knockout result in a dominant negative for slack?), and why does the western blot of knockout show a complete lack of protein rather than a reduction in protein size (does the knockout cause a frame-shift)?

8) It would be informative if the authors address the significant depolarization in resting membrane potential in the knockout. Does this imply that slack channels are open at rest rather than during the rise to threshold?

9) A discussion regarding the discrepancies in the effect of the knockouts on firing patterns between Lu et al., and this study, would also be valuable.

---

## [Author Response]

As explained in detail below, a number of additional experiments need to be included to rule out alternative interpretations of the results. We feel that these can be reasonably accomplished.

*There are several major issues in the present paper concerning the interpretation of the electrophysiology experiments. In the first series of experiments, the authors show that a Na-dependent conductance is produced by cell dialysis with 70 mM and that this conductance is lacking in the* slo2.2*-KO animals. In this part of the paper, everything suggests that this conductance is completely lacking in WT cells dialyzed with 0 mM Na. But in the second section of electrophysiology, the authors examine changes in firing behavior in the KO animals and find that the changes in excitability are exactly the same when the cells are dialyzed with 0 mM Na as with 10 mM Na. They then interpret the effects in 0 mM Na as suggesting either that the* slo2.2 *conductance can actually be present even in the absence of internal Na, or that there is somehow enough Na-entry even at subthreshold voltages to activate the channels. The first possibility seems completely at odds with the first series of experiments, where they see no difference between WT and KO animals with 0 mM internal Na. This seeming inconsistency is not discussed. And there is no attempt to actually test the second possibility.*

*The authors never consider another possibility – that the changes in excitability are not directly due to loss of Na-activated K current but rather reflect changes in other conductances occurring as a result of developmental changes occurring in the KO animals, or other effects involving changes in protein synthesis by loss of* slo2.2*, for which there is previous evidence. This seems likely. As a whole, the changes in excitability seem far more consistent with a change in other conductances (e.g. an increase or change in voltage-dependence of voltage-activated sodium current, or possibly a reduction in a voltage-dependent potassium current) than a loss of Na-activated K current.*

We thank the reviewers for raising the above issues. It was certainly not any intent of ours to overlook any possible explanations for differences in excitability between WT and dKO cells and we hope the changes we have made remedy this issue. The main point we had sought to make regarding the changes in excitability was that the effect on rheobase requires a difference between WT and KO cells in something preceding the AP. Although we would still suggest that, given that we demonstrate that K_Na_ is absent in the KO cells, the absence of K_Na_ current is the simplest potential explanation for such a difference, we have now spent more time, both in new experiments and in discussion, addressing other possibilities, including possible compensatory changes in other currents.

We also thank the reviewers for better focusing attention on the question of the circumstances of when K_Na_ is active and what our results with 0 and 10 mM pipette Na have to say about this issue. The crux of this issue concerns both when K_Na_ current is active and how any activation is controlled by Na^+^. Although we agree that a full answer to these questions is critical, it is beyond the scope of this paper and actually represent questions that have been posed and left unanswered for well over 20 years now. However, we have brought some new data to bear on this topic. We have included new results that more strongly argue that there is no basal activation of K_Na_ in WT neurons with 0 mM pipette Na^+^ (Figure 9—figure supplement 2). Furthermore, over the voltage range of -120 to -60 mV, there is no clear difference in input resistance between WT and dKO cells and, importantly, Cs^+^ does not reduce the resting conductance, negative to -60 mV. But we do consistently observe a difference in resting potential between WT and dKO neurons, which requires some difference in conductance in voltage range between -60 and -45 mV. Although our results do not allow definitive identification of currents active in this range, two factors lead to the hypothesis that a small amount of K_Na_ is active near resting potentials: first, that we know that K_Na_ is absent in the KO and, second, the resting potential of the DRG neurons is fairly positive. Since neither 0 nor 10 mM Na is thought to be sufficient to produce such activation, part of this hypothesis requires that there be some basal Na flux that produces a local [Na] sufficient to produce the very low Po K_Na_ activity required to produce the modest hyperpolarization in WT cells. We also include some simple calculations regarding how much K_Na_ activation would actually be required at rest. Although such considerations are admittedly rather speculative, they do highlight how very small conductances can play very important roles near resting potentials.

We would agree that it would be preferable that such questions be resolved and not simply be presented as a hypothesis. However, this hypothesis will require considerable work and, given the small amplitude of the currents likely to be involved at rest and the absence of selective pharmacological agents, it will be a challenging task. However, we feel the present results provide rather strong support for the idea that K_Na_ is active at a time during AP generation that was not proposed in earlier work and the results at point the way towards strategies that might be useful for more explicit demonstration of time course and amplitude of K_Na_ activation.

The implication of the lack of difference in firing between 0 and 10 mM Na^+^ is simply that these levels of basal Na, which apparently do not activate K_Na_ current themselves, also do not produce much discernible influence on whatever the activating Na actually is. As with other aspects of the results, this would tend to suggest that K_Na_ activation is intimately connected to Na influx.

We also agree that some consideration of the possibility of compensatory changes in other conductances in the KO animals is required and this is now included. In our new data (Figure 9 and supplements), two observations support the view that changes in Nav gating range are not the explanation for the greater excitability of dKO neurons: first, simple inhibition of the ramp-activated outward current in WT cells by the concentration of Cs found to largely block K_Na_ current produces a shift in the 0 current membrane potential very similar to that observed in dKO neurons. In contrast, Cs produces a minor shift in the dKO neurons. Thus, simple alteration of K^+^ conductance preceding the surge of inward current can shift the apparent (but not real) range of Nav activation. Furthermore, in Figure 9—figure supplement 2 we show that there is difference in the range of activation of Nav current between WT and dKO cells that would account for the observed shifts in 0 current potential and rheobase. Similarly, there are no changes in the major voltage-dependent K^+^ current that might account for the excitability changes. Given the heterogeneity of current phenotypes in DRG cells, we think additional work with fully isolated currents will be appropriate, but for this set of cells which reveals the same properties of ramp-activated outward current differences between WT and KO cells as described in Figure 8, we can recapitulate essentially the main features of dKO by simple inhibition of outward current preceding the AP.

We are not so persuaded that the “changes in excitability seem far more consistent with a change in other conductances”. In fact, the effects on firing are generally consistent with what one would expect from an absence of SLO2 and, given that SLO2 is the molecular entity being deleted, that the change in firing results from its absence would certainly seem the simplest explanation for the observations. But as the review of the original manuscript correctly points out, it is absolutely the case that this is not the only possible explanation. However, we hope that our discussion of how compensatory changes might impact on our results now provides a more balanced presentation of the possible interpretations of the results and will draw readers’ attention to the complexity of effects involved with human SLO2.2 mutations.

Finally, in doing this work, a major concern of ours has been that our conclusions from electrophysiological experiments are based on comparisons between WT and KO cells and do not involve within cell comparisons. We feel the new experiments with Cs^+^ provide in-cell comparisons that strongly bolster the arguments derived from WT vs. KO comparisons. However, it may be wondered why we choose Cs^+^ for these experiments. We do, in fact, have a set of experiments with various compounds thought to act on SLO2 channels, both activators and inhibitors, with which we hoped that we might obtain within cell comparisons that would either further support or contradict the comparisons between KO and WT cells. Unfortunately, we obtained very unsatisfying results with loxapine, bithionol, and bepridil, all three of which produce inhibition of Nav current at concentrations comparable to those affecting K_Na_ channels. The slow onset of action of such compounds was also problematic. Although Cs^+^ cannot be considered specific in its blocking effects, the rapidity of onset of its action was more desirable in terms of knowing the effects were due to Cs as opposed to some sort of slow run-down.

*1) As noted, there is a fundamental contradiction between the experiments in*
Figure 6
*showing that the conductance identified as* slo2.2 *current is lacking when cells are dialyzed with 0 mM Na and the interpretation of the experiments in*
Figures 7 and 8
*where it is assumed that this conductance is somehow present even with 0 mM internal Na. This discrepancy needs to be addressed. In*
Figure 6*, there is no data for the double KO with 0 mM Na internal. If the conductance with this condition is really the same as WT with 0 Na, then the speculation later that there might be Na-independent activity of the* slo2.2 *conductance cannot be true.*

In the original Figure 6, current estimates from WT with 0 Na were compared to WT with 70 Na and dKO with 70 Na. Since the K gradients are totally different, we have removed the WT with 0 Na from this panel, but now include WT at 70 mM following 30 minutes of recording, during which time the K_Na_ current runs-down (also shown in new Figure 6—figure supplement 1). To address more directly the question of K_Na_ current at 0 Na, in the new set of cells we measure conductance from -120 to -60 mV. Although a smaller conductance was observed in the dKO cells, neither the conductance in WT or dKO cells was reduced by 20 mM Cs, so we the overall results support the view that there is no basal K_Na_ activation negative to -60 mV. We certainly did not intend to “assume” that K_Na_ was present with 0 internal Na, but this seemed one possible hypothesis to be considered. The current data now allow a stronger conclusion that there probably is no Na-independent activity of SLO2.2.

*2) Also inconsistent with the possibility of a Na-independent conductance from* slo2.2 *channels is that the input resistance is not changed between WT and KO animals. (What was the protocol for measuring input resistance? From what voltage? Hyperpolarizxing or depolarizing current steps? How big? All of these are relevant to interpretation.)*

We have now included the details for measurement of input resistance. In the data in the Tables, we used a 10 mV step from -60 to -70 mV. We think the failure to observe a clear difference in input resistance between WT and dKO cells with this protocol arises from the small amplitude of the current injection, and that negative to -60 mV, there isn’t much difference. We confirmed this with the new data set measuring input resistance from -120 to -60 mV.

*3) The changes in excitability seem more consistent with an increase in voltage-dependent sodium current near threshold – possibly a shift in voltage-dependence to more negative voltages – than a change in potassium conductance. In the data in*
Figure 8*, the initial outward current in the KO seems similar to control, but then the inward current seems to activate at more negative voltages. A shift in the voltage-dependent of sodium current is also the most obvious explanation for the changes in in the phase-plane plot in*
Figure 8*. The authors need to test whether there is a change in inward sodium current in the knockout animals by recording with K current blocked.*

These issues were addressed in our above summary. We would be hesitant to agree with the view that the “changes in excitability seem more consistent with an increase in voltage-dependent sodium current near threshold” based on the results in Figure 8. Our results provide abundant direct evidence that an outward current with the potential to be active near rest (K_Na_) is clearly absent in the dKO cells. Any other differences, if they occur at all, are not readily discernible. Both in Figure 8, the observed currents represent a composite of both inward and outward and any apparent shift in voltage-dependent Nav activation may only reflect the absence of an outward current. But the new experiments with Cs^+^ provide more direct support that a reduction in K^+^ at subthreshold potentials can produce essentially the same effect as SLO2 KO. Furthermore, we see no change in inward sodium current that could account for the changes in excitability.

The review comments suggest that we undertake a comparison of Nav currents in WT and KO cells with all K^+^ current blocked. We agree this may perhaps be an important study to undertake, but feel it is beyond the scope of the present paper. The choice we have taken in using Cs was to allow us to examine the effects of the blocker (Cs) not only on the underlying currents but also on the ability of the cell to fire in the rheobase measurements. We think this provides a more direct test that the changes in current we see can account for the observed excitability changes.

*4) A change in non-voltage dependent potassium conductance might shift the frequency-current curves in*
Figure 6
*to the right, but there is no obvious reason it would result in an increased maximal frequency of firing. This also seems more easily explained by a change in sodium current. Obviously, a change in sodium current would also explain why the effect on excitability are the same with 0 internal and 10 mM internal Na.*

Our description of Nav as a largely “voltage-independent” K^+^ conductance was imprecise and we have now clarified this in the text. We did not mean that KNa_Na_ functions as a voltage-independent current. When cells are loaded with 70 mM pipette Na^+^, the current appears to be largely voltage-independent. But at a fixed and elevated Na, some of the linearity of K_Na_ activation over then range of -80 to -20 mV will arise from a combination of increased activation by voltage and also voltage-dependent inhibition by intracellular Na. Furthermore, under more physiological conditions, it would be expected that K_Na_ will increase in activation with depolarization weakly from its own intrinsic voltage-dependence, but more strongly as a consequence to elevations of cytosolic Na. We have to admit that it is not so obvious to us how to predict effects of a current active prior to an AP on the frequency-current plots, but we understand the basic point raised by the reviewer.

*5) Another possibility is a loss of a voltage-dependent K current. In the IV curves in*
Figure 6
*shown for the dKO animals with 70 mM internal Na, it is striking that there is no evidence of any voltage-dependent component of K current like that seen for the WT neurons with 0 mM internal Na.*

Thank you for focusing attention on this feature of the IV curves. The beginning of outward current activation in the WT cells at 0 Na in fact reflects K_Na_ activation in those cells. This current runs down over time consistent with our other observations on run-down of K_Na_ current. This provides support for the idea that under more physiological conditions this current does exhibit voltage-dependent activation.

*6) More generally, the authors interpret all of their results as if they represent direct effects of a loss of* slo2.2 *rather than possible developmental changes. Such changes seem not only possible but likely based on the fact that* slo2.2 *mutations in humans seem to have profound developmental effects and with previous evidence that* slo2.2 *protein can affect synthesis of other proteins. To quote the final summary of the 2013 review by Kaczmarek, “Human mutations in these channels produce profound effects on neuronal function and development, suggesting that perhaps their biological role may extend beyond simply setting the level of neuronal excitability, and that these channels may influence cytoplasmic biochemical pathways that regulate development, plasticity, and intellectual function.” Obviously, such indirect effects could offer a completely different explanation of the changes in excitability – or even the effects on pain and itch behavior. This needs to be acknowledged and considered in the interpretation in this manuscript.*

We have now included discussion of the possibility that the absence of *Slo2* protein may affect expression of other proteins. The hypothesis cited by the review comment is an intriguing one that will certainly require additional evaluation.

7) Because this is the first description of these knockouts, the authors should provide some rationale for gene targeting of the particular exons that were selected (perhaps in the Methods?). The authors should address what the selected exons encode, if knockout of the exon conceivably causes a dominant negative protein for its paralogue channel (i.e., is slick exon knockout result in a dominant negative for slack?), and why does the western blot of knockout show a complete lack of protein rather than a reduction in protein size (does the knockout cause a frame-shift)?

We have expanded on these points in the Methods.

8) It would be informative if the authors address the significant depolarization in resting membrane potential in the knockout. Does this imply that slack channels are open at rest rather than during the rise to threshold?

This was largely addressed above, but is repeated here. Our results lead us to suggest that slack channels are open at rest, at very low open probability, but that slack activation also increases during the rise to threshold. As discussed more thoroughly above, we have more explicitly addressed this issue both with new experiments and in the Discussion. However, this is not an easy question to definitely answer, and the identity and magnitude of the full set of conductances active at rest (and subthreshold) are not readily identified. As noted above, the discrepancy our results raise is that, although no differences in input resistance (measured negative to -60 mV) are noted between WT and dKO cells, we have consistently observed resting potential differences that do not seem readily explained solely on the basis the unfortunate heterogeneity of DRG neurons. Given that the relatively linear IV from -60 to -120 mV begins to markedly deviate above -60 mV, SLO2 channels are certainly candidates for a conductance that might contribute in this window. However, as we discuss, such a conductance would be very small at rest and activation at rest, but not negative to -60 mV would seem likely to require coupling to a source of Na influx in this window. Unfortunately, that remains a topic about which we know essentially nothing, so although the results suggest particular explanations, they remain decidedly speculative. Although it is to be expected that there is a Na flux in the -60 mV to -50 mV window that contributes to the relatively positive resting potentials, whether the coupling could be strong enough to produce even small K_Na_ activation is unknown. These complicated questions are not easy to tease apart, given the small size of the conductances/currents and woefully nonspecific pharmacological agents. We think the present work points to some approaches that might be taken to address this more definitely in the future work. It was not the purpose of this manuscript to explain fully the role of K_Na_ current during DRG electrical activity although that would have been nice. However, the effects on rheobase strongly focused our attention on processes that precede the upswing of the AP. For now, we feel the approaches we have taken are important in establishing a new way that Slack current can influence excitability and point to important new questions.

9) A discussion regarding the discrepancies in the effect of the knockouts on firing patterns between Lu et al., and this study, would also be valuable.

An additional comment in regards to the firing patterns is now included. However, the Lu paper does not undertake an examination of the basis for any firing differences in their KO animals, the size of neurons from which they record was not reported, and we simply felt there was no need to point out things that might be lacking in the other paper. The key point is that both papers note that excitability in response to constant current injection is increased. We feel that this is consistent with the idea that acute effects on sensory stimuli would likely be observed, even in the absence of neuropathic pain protocols.